# Scaling Agents for Computer Use

## Abstract

Computer-use agents (CUAs) hold promise for automating everyday digital tasks, but their performance on long-horizon, complex problems remains unreliable. Single-rollout execution is brittle, with small errors compounding over time and leading to high variance in outcomes. While prior work has attempted to scale within a single rollout, such approaches have yielded limited gains. Scaling over multiple rollouts offers a more promising alternative but doing so effectively is challenging due to the difficulty of evaluating and selecting among long-horizon agent behaviors. We introduce *Behavior Judge* (BJudge), which addresses this challenge by representing agent executions as behavior narratives and comparing candidate behaviors at this level, substantially improving robustness and success rates. Using multiple rollouts, BJudge establishes a new state of the art (SoTA) in OSWorld at 72.6%, outperforming the previous best by 9.2% and performing comparably to human-level performance (72.36%), with comprehensive ablations validating key design choices. We further demonstrate promising cross-platform transfer to different operating systems on WindowsAgentArena and Android-World. Crucially, our results highlight the strong effectiveness of scaling CUAs, when you do it right: effective scaling requires structured trajectory understanding and selection, and BJudge provides a practical framework to achieve this.

## 1 Introduction

Computer-use agents (CUAs) offer the promise of automating everyday digital tasks across operating systems and applications (Xie et al., 2024; Song et al., 2025; Guo et al., 2025b; Yang et al., 2025b; Xie et al., 2025; Wang et al., 2025a;b). Yet despite rapid advances, current CUAs remain unreliable on long-horizon, complex problems. The difficulty lies not only in solving individual steps but also in sustaining correctness across dozens or even hundreds of interactions. Small mistakes accumulate, feedback is often delayed, solution paths branch in unpredictable ways, and environmental noise (UI changes, pop-ups, latency) further destabilizes performance (Yang et al., 2025a). Together, these factors lead to high variance in outcomes, with the same agent often succeeding on one attempt but failing catastrophically on another.

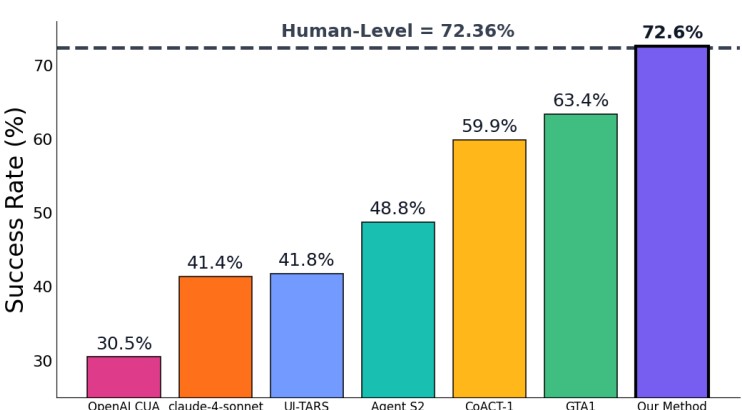

Figure 1: Performance on OSWorld at 100 steps. Our method beats the previous SoTA by 9.2% absolute improvement and performs comparably to human-level performance on OSWorld.

A natural response to this variance is to scale execution. Prior work (Yang et al., 2025b) has explored scaling within a single rollout to combat variance but single-trajectory scaling has led to limited improvements in overall success on long-horizon tasks. An alternative approach is *wide scaling*: instead of simply accepting a

Figure 2: Disjoint task success across rollouts by three agent instances. Behavior Judge (BJudge) leverages this complementarity by selecting the best trajectory among multiple rollouts.

single rollout from one agent, we can scale the number of agents to generate multiple rollouts in parallel and select the best. This wide scaling perspective leverages the fact that agents, while suboptimal individually, often succeed on complementary subsets of tasks, as shown in Figure 2.

In practice, however, *wide scaling exposes a fundamental bottleneck: evaluation.* While generating multiple rollouts is straightforward, reliably determining which long-horizon trajectory is correct is not. Agent trajectories are information-dense and multimodal, with most details irrelevant to task success, making them difficult to interpret, compare, and evaluate. Moreover, many computer-use tasks admit multiple valid solutions, and scripted automatic evaluation struggles to decide whether a trajectory is correct (Xie et al., 2024; Rawles et al., 2025; Bonatti et al., 2025).

To address these challenges, we introduce Behavior Judge (BJudge), a novel framework that enables wide scaling of CUAs through principled trajectory representation and selection. Our approach first converts raw trajectories into behavior narratives: concise summaries that capture what the agent actually did and how it affected the environment, preserving task-relevant action–effect summaries while filtering away irrelevant detail at individual steps. These narratives provide a compact yet faithful representation that makes it easier for a judge to compare candidates. BJudge then performs selection directly over narratives, enabling reliable selection among multiple rollouts. In addition, we build upon existing CUAs and introduce an improved computer-use agentic framework to generate high quality trajectories for BJudge.

Our method delivers strong performance on computer-use benchmarks. On OSWorld (Xie et al., 2024), it achieves a new state of the art with a 72.6% success rate (100 steps), surpassing the previous best of 63.4% and performing comparably to human-level performance at 72.36% (Figure 1). Beyond OSWorld, our approach also shows promising zero-shot cross-platform transfer to WindowsAgentArena (Bonatti et al., 2025) and AndroidWorld (Rawles et al., 2025).

Our contributions are four-fold:

- We introduce the wide scaling paradigm for CUAs, showing that carefully selecting from multiple trajectories substantially improves robustness and coverage.

- We propose Behavior Judge (BJudge), a framework that enables the evaluation of multiple rollouts by converting dense trajectories into compact behavior narratives and using them for principled trajectory selection.

- Our method, together with an improved CUA baseline, achieves a new SoTA of 72.6% on OSWorld, surpassing prior work by a large margin (9.2% absolute improvement).

- We provide extensive ablations validating our design choices and demonstrate promising zero-shot cross-platform transfer to WindowsAgentArena and AndroidWorld.

## 2 Background

### 2.1 Computer-Use Agents

Computer-use agents (CUAs) executing user instructions can be framed as a partially observable Markov Decision Process (POMDP) defined as $\mathcal{M} = \langle \mathcal{S}, \mathcal{O}, \mathcal{A}, \mathcal{T}, \mathcal{I}, R \rangle$, where $\mathcal{S}$ is the state space encoding the computer state, $\mathcal{O}$ is the observation space such as desktop screenshots, $\mathcal{A}$ is the action space of the agent (e.g. `agent.click(...)` and `agent.type(...)`), $\mathcal{T} : \mathcal{S} \times \mathcal{A} \to \Delta(\mathcal{S})$ is a stochastic transition function, $\mathcal{I}$

is the space of possible user instructions represented in natural language, and $R : (\mathcal{S} \times \mathcal{A})^* \times \mathcal{I} \to [0, 1]$ denotes the instruction reward function that assigns a scalar reward to a trajectory of states and actions $\tau := (s_0, a_0, \ldots, a_{T-1}, s_t)$ on task $I$. We use $h_t := (o_0, a_0, \ldots, o_{t-1}, a_{t-1}, o_t)$ to denote a time-ordered history of all consecutive observations and actions up to and including $o_t$.

A broad spectrum of computer agents has been explored including general agentic frameworks (Song et al., 2025; Yang et al., 2025b; Agashe et al., 2025b;a), generalist agents (Anthropic, 2025; OpenAI, 2024; Guo et al., 2025a) and graphical user interface (GUI) agents (Wang et al., 2025a; Xu et al., 2025). These prior work consider a single model as the policy $\pi(a|h_t, I)$ that, when executed, yields one trajectory $\tau = (o_0, a_0, \ldots, o_T)$ where $a_t \sim \pi(\cdot|h_t, I)$. In contrast, our work is the first, to our knowledge, that focuses on scaling the number of candidate solution trajectories by using multiple base models and policies, and we propose effective methods to select the optimal solution.

## 2.2 Test-Time Scaling

A common strategy for improving large multimodal models and their agentic extensions is through test-time scaling (Zhu et al., 2025), where multiple solutions are generated either in parallel or sequentially, followed by selection of a final response using a reward model or iterative generation of new solutions (Snell et al., 2025; Lightman et al., 2024; Jain et al., 2025). Recent work (Yang et al., 2025b) has adapted this idea for CUAs with *step-wise BoN* (Zhu et al., 2025), where at each step the agent $\pi$ generates $K$ candidate actions $\mathcal{C}_t = \{a_t^{(k)}\}_{k=1}^{K} \sim \pi(\cdot|h_t, I)$ and then a judge $J$ selects the best action $\hat{a} = J(\mathcal{C}_t)$. While this can help with local improvements, it commits the rollout to the current agent plan. In tasks with multiple valid solutions paths, this can lead the agent to over-commit to a harder route, missing easier alternatives. In contrast, our work investigates the *wide scaling* approach where a judge must select the best trajectory among a set of candidate trajectories generated by multiple base agents or models. Wide scaling can be viewed as a trajectory-level analogue of best-of-N sampling, but unlike standard best-of-N over i.i.d. samples from a single model, our candidates are full long-horizon trajectories from multiple agents.

However, wide scaling is non-trivial because trajectory evaluation is still a fundamental challenge. Most existing benchmarks such as OSWorld (Xie et al., 2024), WindowsAgentArena (Bonatti et al., 2025), and AndroidWorld (Rawles et al., 2025) use evaluation scripts written by humans which cannot be scaled. In contrast, work on web-agent benchmarks, a subset of CUA focused on browsers, has explored using vision-language models (VLMs) as judges (He et al., 2024; Deng et al., 2023; Xue et al., 2025). However, these judges are typically tuned for the web domain, require human-defined rubrics, and do not generalize well to the broader tasks faced by CUAs. In addition, aligning such judges with human judgment requires substantial manual effort, such as in Mind2Web 2 (Gou et al., 2025) that achieved 99% agreement using code-generated rubrics but still relied on extensive human verification. Moreover, all these evaluation methods only work with a single trajectory. Our work aims to implement a judge to handle trajectory evaluation by (1) improving trajectory understanding by converting trajectories into a behavior narrative that describes what an agent did and (2) comparing trajectories using the behavior narratives to effectively distinguish the best.

## 3 Method

Our **Behavior Judge** framework, shown in Figure 3, is designed to enable wide scaling over many agent rollouts. We improve upon Agent S2 (Agashe et al., 2025b), a top-performing open-source agentic framework, and introduce two key components: *Behavior Narrative Generator* and *Comparative Behavior Evaluator*. Given a rollout, the Behavior Narrative Generator derives facts from each transition, yielding a behavior narrative that describes what the agent did (action-effects) while discarding irrelevant details. The Comparative Behavior Evaluator then conducts a comparative evaluation of the candidate narratives across multiple rollouts to determine the best solution.

### 3.1 Behavior Narrative Generation

Long-horizon trajectories are information-dense, with every step producing a new screenshot. We argue that it is not necessary or even optimal to judge all of the raw visual content directly to understand what actually

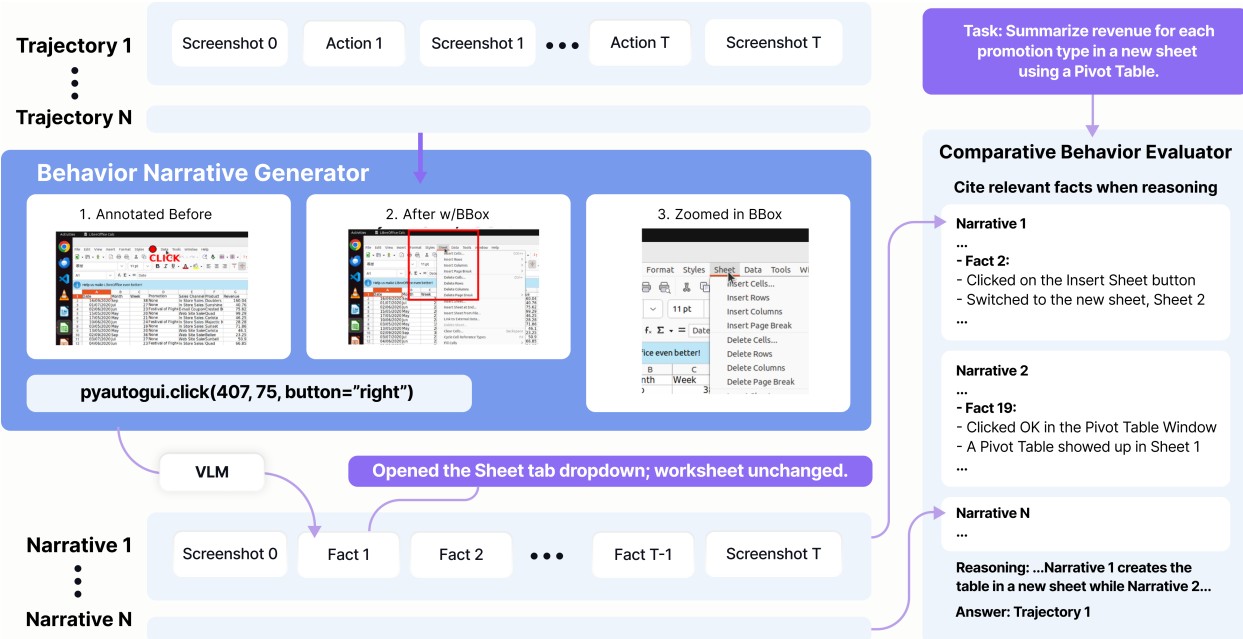

Figure 3: Behavior Judge generates multiple rollouts consisting of screenshots and actions. These trajectories are converted into behavior narratives via the Behavior Narrative Generator, using the executed action and before/after screenshots to describe what was changed. Finally, the behavior narratives are provided to the Comparative Behavior Evaluator which selects the best trajectory through comparison.

occurred. We propose to extract the task-relevant changes caused by the agent's actions from screenshots in order for a downstream judge to focus on the changes that matter. We construct a behavior narrative composed of facts that describe what the agent did at each step. Concretely, given a generator $G$ (instantiated using a VLM) and an agent rollout $\tau = (s_0, a_1, s_1, \ldots, a_{T-1}, s_T)$ where $s$ denotes a screenshot and $a$ denotes an agent action, we feed in transitions $(s_i, a_i, s_{i+1})$ to the generator and derive facts $\phi_i = G(s_i, a_i, s_{i+1})$, for each $i \in \{0, \ldots, T-1\}$.

To generate accurate facts, the Behavior Narrative Generator takes in a screenshot before action execution, the action to execute, and the screenshot after execution as depicted in Figure 3. The generator applies targeted visual augmentations for pointer interactions (clicks, taps, moves, and drags), as these actions require pixel-level precision and are more prone to agent hallucination. For example, a step-level hallucination where a click on the Save button fails but the agent believes otherwise can be the difference between a success or failure. On the screenshot before action execution $s_i$, we overlay a marker centered at the pointer coordinate $(x_i, y_i)$ where $a_i$ will occur. On the screenshot after action execution $s_{i+1}$, we extract a zoomed crop $s_{i+1}^z$ of a fixed-size square centered at the final pointer coordinate $(x_{i+1}, y_{i+1})$ and outline the crop in $s_{i+1}$ to indicate the region of interest. The zoom provides the generator with fine-grained evidence to verify that the intended change occurred. To handle cases where changes are delayed (e.g. clicking a hyperlink), screenshot $s_{i+1}$ is taken 3 seconds after action execution.

Once facts have been derived from each transition, we construct a behavior narrative $\tilde{\tau} = (s_0, \phi_0, \phi_1 \ldots, \phi_{T-1}, s_T)$ that retains only task-relevant changes. We include the initial and final screenshot to ground where changes begin from and what they result in. This allows Behavior Judge to focus solely on what the agent did differently between trajectories. We provide a qualitative example in Appendix N.

## 3.2 Comparative Behavior Evaluator

While generating multiple rollouts increases the chance that at least one rollout is successful, the benefits can only be realized if we can reliably select the correct trajectory. Selection is challenging because an evaluator

must both interpret long-horizon behavior within each rollout (to verify task requirements) and discriminate among candidates. To simplify this, we decide to separate these responsibilities by generating a concise behavior narrative that describes the long-horizon behavior so the bulk of the evaluator's responsibility lies on selecting between candidates.

Concretely, given a set of base policies $\{\pi_m\}_{m=1}^M$, we generate candidates $\mathcal{C} = \bigcup_{m=1}^M \{\tau_m^{(n)}\}_{n=1}^{N_m}$ where each candidate $\tau_m^{(n)}$ is sampled via stochastic decoding from a base policy $\pi_m$. This allows us to capture diversity from variance within the same model ($n = 1 \ldots N_m$) and differing capabilities across different models ($m = 1 \ldots M$). Our objective is to select the candidate trajectory that maximizes task return $\hat{\tau} \in \arg\max_{\tau \in \mathcal{C}} R(\tau, I)$. The candidate set $\mathcal{C}$ is converted to a corresponding set of behavior narrative candidates $\tilde{\mathcal{C}} := \{\tilde{\tau}^{(n)}\}_{n=1}^{|C|}$, according to the behavior narrative generation in Section 3.1. Then a VLM evaluator $E$ is prompted to run comparative evaluation using all narratives in $\tilde{\mathcal{C}}$ and select a single best narrative candidate, which corresponds to the final selected trajectory $\hat{\tau} \in \mathcal{C}$. In this work, we instantiate comparative evaluation using a single-round multiple-choice question (MCQ) format, which enables a more informed comparison than independent ranking while being more token-efficient and faster than multi-round tournament-style comparisons of subsets of candidates. The system prompt (Section M) emphasizes on citing and contrasting facts to ensure each candidates' behaviors are carefully observed, which we find gives small improvements (Section K). By comparing behavior narratives altogether, we enable wide scaling over many agents.

### 3.3 An Improved Agentic Framework Baseline

As Behavior Judge operates on multiple full-length trajectories generated by base agents, we can improve the overall performance and latency of BJudge by starting with the best frameworks for the base agents. Inspired by Agashe et al. (2025b) and Song et al. (2025), we created an improved baseline agentic framework, *Agent S3*, which achieves strong performance even before incorporation into BJudge. It draws upon two key ideas: 1) performance gains of programmatic edits over direct GUI manipulation when needed (up to the agent itself), and 2) speedup by using a flat (worker only) policy instead of a manager-worker hierarchy.

**Coding Agent** To encourage diverse solution paths, our GUI policy $\pi(a_t \mid I, h_t)$ reasons what approach might be best suited for the next step: generate a GUI action $a_t \in \mathcal{A}_{\text{gui}}$ or invoke the *coding agent* for programmatic edits (e.g., bulk operations, file transforms, structured parsing). A code call launches a bounded inner loop with budget $B$ that iterates on generated code and terminal *feedback*. At inner step $k$, the coding agent conditions on $c_k^{\text{code}} = (I, o_t, F_{1:k-1})$, where $F_{1:k-1}$ aggregates execution signals (status, return code, stdout/stderr) from prior iterations. It either emits Python/Bash to be executed in a sandboxed VM, or returns a control token `DONE`/`FAIL`. On termination, a brief summary of the session—logic, observed effects, and a verifiable inspection checklist—is appended to the GUI agent's history to aid on-screen verification and subsequent planning by the GUI policy. Different from Song et al. (2025), our coding agent implementation does not use the AutoGen Wu et al. (2024) framework nor does it use an orchestrator to divide and delegate tasks across the GUI and coding agents. Our coding agent implementation is natively integrated into our GUI agent's action space, allowing GUI agent to reason when best to delegate the next step to the coding agent.

**Flat Policy** We remove hierarchical planning in favor of a flat policy that can replan at any time based on $(I, h_t)$. Contemporary foundation models exhibit strong GUI understanding and can maintain short-horizon plans in context, making a separate high-level planner unnecessary and sometimes counterproductive (e.g., when subgoals become stale). We evaluate these design choices in Table 2; implementation details appear in Section I.

## 4 Experiments and Analysis

In the following experiments, we systematically investigate the effectiveness of Behavior Judge (BJudge) across several dimensions of computer-use agents. Specifically, we aim to address the following research questions:

1. **Performance.** How does BJudge perform compared with other CUA baselines? (Section 4.2)

2. **Equal rollout budget.** How do judging strategies perform under an equal rollout budget? (Section 4.3)

3. **Representation.** How do behavior narratives compare to other trajectory representations? (Section 4.4)

4. **Failure modes.** How accurate is BJudge's evaluation and what are its main failure modes? (Section 4.5)

5. **Resource allocation**. How does BJudge perform with varying steps across workers? (Section 4.6)

6. **Ensembling.** How should we select a mixture-of-models ensemble for rollouts? (Section 4.7)

7. **Generalizability.** How does BJudge generalize to other domains and benchmarks? (Section 4.8)

## 4.1 Experimental Setup

**Benchmarks**   We focus on *OSWorld* (Xie et al., 2024), which comprises 369 real-world Ubuntu tasks across five domains (OS, Office, Daily, Professional, Workflow). Following common practice (Xie & et al., 2024), we use the 361-task subset that omits eight multi-application tasks requiring Google Drive credentials not available in the sandbox. We further assess generality beyond Ubuntu on two additional benchmarks: *WindowsAgentArena* (Bonatti et al., 2025), a 154-task Windows benchmark, spanning LibreOffice Writer/Calc, Edge/Chrome, File Explorer/Windows Settings, VS Code, VLC, and utilities; and *AndroidWorld* (Rawles et al., 2025), a 116-task Android benchmark with step budgets specified by the benchmark authors.[1]

**Baselines**   On OSWorld, we introduce Agent S3 as an improved baseline for scaling results. We additionally compare against other top methods including Jedi (Xie et al., 2025), GTA1 (Yang et al., 2025b) and CoACT-1 (Song et al., 2025). For AndroidWorld, we compare with 3 top-performing open-source frameworks using screen-shot only representations including MobileUse (Li et al., 2025), UI-Venus (Gu et al., 2025), and Agent S2 (Agashe et al., 2025b). For WindowsAgentArena, we compare with Navi (Bonatti et al., 2025) and Agent S2 (Agashe et al., 2025b). For ablation of the judge for scaling, we compare against an adaptation of WebJudge (Xue et al., 2025), which has 85% agreement with human judgment, for isolating the effect of comparative versus independent trajectory selection mechanisms. We also implement and compare against three baselines when isolating the effect of representation: 1) a naive captioner that captions each screenshot individually, 2) a trajectory summarizer and 3) using screenshots only.

**Implementation Details**   Agent S3 is an improvement over Agent S2 that removes hierarchical planning and adds a coding agent (details in Appendix I). We use Agent S3 to generate rollouts for BJudge trajectory selection. The coding agent is enabled for OSWorld and WindowsAgentArena but disabled for AndroidWorld due to emulator constraints that preclude program execution and inspection. Our best score uses 5 rollouts from GPT-5 and Opus 4.5 each and uses Opus to generate facts while GPT-5 selects the best rollout. We also adapt WebJudge to do comparative selection by individually ranking each trajectory with a score 1-5 and choosing the highest score, tie-breaking at random, and we adapted the system prompt to the OS setting. For our Screenshot Only baseline, we pass $50/N$ screenshots per trajectory chosen at uniform intervals across the trajectory, due to context length limitations. Details on cost and runtime can be found in Appendix B, and full reproducibility details in Appendix G. Unless otherwise noted, we report 95% confidence intervals from a nonparametric bootstrap over the 361 tasks (10,000 resamples), paired over per-task score differences when comparing methods, and report non-scaling baselines as the mean $\pm$ standard error over 10 independent runs (further variability analyses in Appendix H).

## 4.2 Main Results

As shown in Table 1, Agent S3 already establishes a strong foundation, achieving strong non-scaling performance on 100-step success rate for OSWorld. Building on this, applying our core contribution, Behavior Judge (BJudge), to multiple rollouts of Agent S3 further improves performance on 100 steps. For example, it achieves 69.9% SR with GPT-5 (a 7.3% absolute improvement; paired bootstrap 95% CI [4.6, 10.0]) and 60.2% SR with GPT-5 Mini (a 10.4% absolute improvement; paired 95% CI [7.3, 14.1]). Our best result at 72.6% (95% CI [68.0, 77.1]) using GPT-5 and Opus 4.5 is comparable to human-level performance (72.36%) (Xie

---

[1]Experiments were conducted under the AndroidWorld step budget guidelines as of September 20, 2025.

| Method | Model | 100-step |
|---|---|---|
| Agent S2 (Agashe et al., 2025b) | GPT-5 | 48.8 |
| Jedi-7B (Xie et al., 2025) | o3 | 51.0 |
| CoAct-1 (Song et al., 2025) | OAI CUA + o3 + o4-mini | 59.9 |
| **Ours** | | |
| Agent S3 | o3 | 61.1 |
| Agent S3 | GPT-5 Mini | 49.8 ± 0.7 |
| Agent S3 | GPT-5 | **62.6 ± 0.5** |
| *Scaling Results* | | |
| GTA1 (step-wise scaling) (Yang et al., 2025b) | o3 | 53.1 |
| GTA1 (step-wise scaling) (Yang et al., 2025b) | GPT-5 | 63.4 |
| Agent S3 w/ WebJudge (N=10) (Xue et al., 2025) | GPT-5 Mini | 50.4 |
| **Ours** | | |
| Agent S2 w/ BJudge (N=10) | GPT-5 † | 56.5 [51.3, 61.6] |
| Agent S3 w/ BJudge (N=10) | GPT-5 Mini | 60.2 [55.1, 65.1] |
| Agent S3 w/ BJudge (N=10) | GPT-5 | 69.9 [65.0, 74.5] |
| Agent S3 w/ BJudge (N=10) | GPT-5 + Opus 4.5 | **72.6 [68.0, 77.1]** |

Table 1: OSWorld success rate (%) on 100-step tasks across 361 tasks. We introduce Agent S3, which achieves strong performance with GPT-5 at 62.6% averaged over 10 runs. Our method, Behavior Judge, enables strong scaling results with new SoTA at 72.6% with an ensemble of GPT-5 and Opus 4.5, chosen from Table 5. For our results, ± denotes the standard error of the mean across 10 independent runs and brackets denote bootstrap 95% CIs over the 361 tasks. † we ran using GPT-5 on Azure which encountered safety filter blocks more frequently (Agent S2 44.3 ± 0.6)

| Method | 100-step SR (%) | LLM calls/task | Time/task (s) |
|---|---|---|---|
| Agent S2 (Agashe et al., 2025b) | 48.8 | 73.62 | 2366.80 |
| Agent S2 (no hier.) | 57.9 (+9.1) | 41.39 (-43.8%) | 1132.91 (-52.1%) |
| Agent S3 | **62.6 (+13.8)** | **35.12 (-52.3%)** | **891.21 (-62.4%)** |

Table 2: OSWorld success rate and efficiency statistics using GPT-5. Baseline is Agent S2 with hierarchical planning; values in parentheses show Δ vs. Agent S2 (for SR and efficiency metrics).

et al., 2024), highlighting that BJudge can boost existing methods to reach human-comparable capability. In terms of cost, generating 10 rollouts averages $7.20 per task while BJudge itself adds only $0.14 for narrative generation and judging (Appendix B); Section 4.6 analyzes the resulting performance-compute trade-off and Appendix C discusses cheaper configurations.

In addition, Table 2 reports the performance and efficiency gains of our improved agentic framework baseline, Agent S3, compared to Agent S2 (Agashe et al., 2025b) that it was built upon. We additionally include Agent S2 w/ BJudge in Table 1 to isolate the improvement of the framework and judge. Agent S3 yields a 13.8% improvement in success rate, a 52.3% reduction in LLM calls per task, and a 62.4% reduction in average task completion time.

### 4.3 How does Behavior Judge perform under equal rollout budgets?

Figure 4 shows a comparison between BJudge and baselines using equal rollout budgets. We modify WebJudge to choose over the same number of rollouts as BJudge by independently ranking rollouts and selecting the highest rank. We find that overall BJudge achieves better performance than baselines when compared equally. We find that WebJudge provides limited benefit over the average performance of rollouts and that BJudge shows better scaling as we increase the number of rollouts. While WebJudge has some slight improvements around N=4, it plateaus quickly and drops around N=10. This suggests that it is necessary to compare trajectories against each other to make selection effective and scalable with any rollout budget.

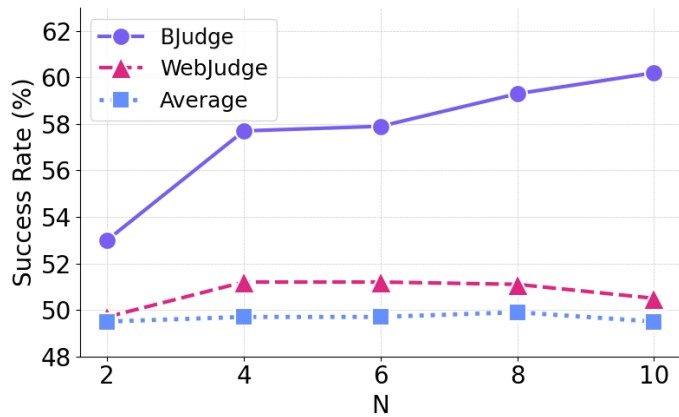

Figure 4: Comparison of BJudge against WebJudge on OS-World using GPT-5 Mini's rollouts. *Average* represents the average performance of the rollouts.

### 4.4 How do behavior narratives compare to other trajectory representations?

Table 3 shows an ablation on our behavior narrative representation. We compare against a screenshot-only baseline, a trajectory summary baseline that summarizes the trajectory in 3-6 sentences, and a naive captioning baseline that captions each screenshot individually. We find that behavior narratives are an effective representation for BJudge, providing a 3.4% improvement over the best baseline. This suggests that it is difficult to understand screenshots alone and that it is necessary to generate facts over transitions rather than individual states.

| Representation | Success Rate (%) |
|---|---|
| Screenshot Only | 56.0 |
| Trajectory Summary | 55.0 |
| Naive Captioning | 56.8 |
| Behavior Narratives | **60.2** |

Table 3: Ablation study on BJudge's behavior narrative representation with 10 GPT-5 Mini rollouts.

### 4.5 BJudge Accuracy and Failure Analysis

| Category | Judge Subset Accuracy | Full Set Accuracy |
|---|---|---|
| Benchmark Alignment | 78.4% [71.7, 84.7] | 69.9% [65.0, 74.5] |
| Human Alignment | 92.8% | 76.3% |

Table 4: BJudge accuracies on Judge Subset and Full Set with 10 GPT-5 rollouts on OSWorld. The Judge Subset consists of a subset of 159 OSWorld problems that could be improved on due to disjoint task success. Brackets denote bootstrap 95% CI over tasks.

Table 4 shows the accuracy of BJudge with respect to OSWorld evaluation scripts and to our human alignment. We find that on 159 problems (Judge Subset) where the judge can improve performance (i.e. where there is at least one correct and one incorrect trajectory), it achieves 78.4% accuracy during selection (95% CI [71.7, 84.7]). After manual inspection (Appendix D) over the remaining 35 problems, we found through human evaluation that the accuracy is 92.8%, as the OSWorld evaluation scripts are imperfect and can only strictly evaluate one pre-defined solution. This suggests that BJudge is highly effective at selecting the right trajectories from multiple candidates.

For the remaining 12 failures, we categorize these as behavior narrative generation hallucinations (8) and Code-GUI handoff failures (4). We observe generation hallucination occur in instances where the underlying VLM has difficulty with visual understanding such as missing fine-grained details in text which zooming has little effect on (e.g. the negative sign on a number as shown in Appendix L). We also observe some cases where the GUI-Agent failed to recognize the Coding Agent's changes, and perform GUI actions overwriting Coding Agent's changes and cause evaluation to fail. These kind of failed rollouts generate rich GUI-related behavioral narratives, which are preferred by BJudge compared to the rollouts whereas the Coding Agent performs everything in one step and completes, outputting limited behavioral narratives. We find this is not due to the actual length of the rollout (Appendix F) but rather due to missing evidence in the screenshots for the behavior narratives to summarize. We compute a domain-wise breakdown in Appendix E.

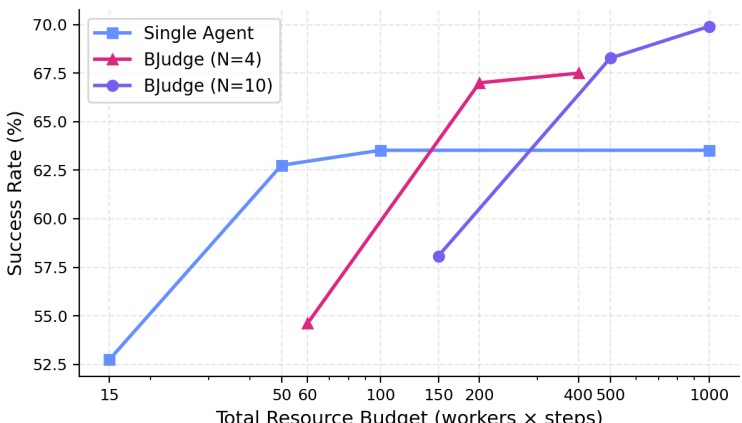

Figure 5: Comparison of BJudge with varying resources calculated by the number of workers times step budget using GPT-5.

### 4.6 How does Behavior Judge perform under varying resource budgets?

Figure 5 analyzes how BJudge performance varies with the total resource budget and the number of workers N. The x-axis reports the total budget, defined as the number of workers multiplied by the per-worker step budget, with each curve corresponding to a different allocation of this budget across workers. At smaller budgets, a single agent performs best, as distributing compute across many workers reduces the per-worker step budget below what is required to complete the task. As the total budget increases, larger values of N become increasingly effective with BJudge (N=4) achieving a 4.25% absolute improvement over a single agent at twice the total budget (200), while the largest gain of 6.38% is observed with BJudge (N=10) at a budget of 1000. These results suggest that moderate values of N offer a favorable trade-off under tighter compute budgets, while larger N yields the highest performance when sufficient resources are available.

### 4.7 How should we select a mixture-of-models ensemble for rollouts?

Table 5 shows the success rate and task coverage of BJudge using various mixture-of-model combinations. Task coverage is calculated by setting a task successful if at least one trajectory is correct, or Pass@N (Chen et al., 2021). We observe that from the single model mixtures, Claude Opus 4.5 performs the strongest at 69.9% followed by Gemini 3 at 67.7%, demonstrating that strong model capabilities lead to overall higher success with selection. We also observe that the most diverse mixture (All) achieves the highest overall task coverage at 80.5%, demonstrating that diversity is key to increasing the upper bound on success. Finally, we observe that the GPT-5 + Claude Opus 4.5 mixture achieves the highest success rate of 71.6%, suggesting that selecting a mixture-of-models ensemble with highly diverse capable models achieves the best performance.

| Mixture | SR (%) | Pass@N (%) |
|---|---|---|
| GPT-5 | 66.5 | 74.7 |
| Claude Opus 4.5 | 69.9 | 74.5 |
| Gemini 3 | 67.7 | 74.3 |
| GPT-5 Mini | 57.0 | 68.2 |
| GPT-5 + Mini | 64.9 | 74.1 |
| GPT-5 + Opus | **71.6** | 79.1 |
| GPT-5 + Gemini | 67.3 | 78.5 |
| Opus + Gemini | 70.6 | 78.2 |
| Opus + Mini | 66.0 | 75.6 |
| Gemini + Mini | 65.1 | 75.3 |
| All | 68.4 | **80.5** |

Table 5: Success rate and task coverage for BJudge (N=4) using mixture-of-model combinations split evenly.

| Method | Model | 50-step | 100-step |
|--------|-------|---------|----------|
| UI-TARS-1.5 | - | 42.1 | - |
| Agent S3 | GPT-5 | 49.0 | 50.2 |
| BJudge (N=3) | GPT-5 | **54.1** | **56.6** |

Table 6: WindowsAgentArena success rate (%) within 50 steps and 100 steps. Behavior Judge (N=3) consistently outperforms the baseline Agent S3, with a 6.4% improvement on 100-step SR.

| Method | Model | SR (%) |
|--------|-------|--------|
| Agent S2 | Claude 3.7 Sonnet | 54.3 |
| MobileUse | Qwen2.5-VL-72B | 62.9 |
| UI-Venus | UI-Venus-Navi-72B | 65.9 |
| Agent S3 | GPT-5 | 68.1 |
| BJudge (N=3) | GPT-5 | **71.6** |

Table 7: AndroidWorld success rate (%). Behavior Judge (N=3) achieves a 3.5% improvement over the baseline Agent S3.

### 4.8 Generalization to Other Benchmarks

Table 6 and 7 show promising cross-platform transfer of BJudge to different operating systems. For AndroidWorld, we compare with top 3 performing open-source, screenshot-only methods including AgentS2 (Agashe et al., 2025b), MobileUse (Li et al., 2025), and UI-Venus (Gu et al., 2025) For WindowsAgentArena, we compare with Agent S2 and UI-TARS-1.5 (Seed, 2025). We find that Behavior Judge, with N = 3, achieves a performance boost of 3.5% and 6.4% respectively, suggesting that our method transfers well across platforms.

## 5 Limitations

Behavior Judge assumes access to an agent capable of producing multiple independent rollouts from the same initial state. This assumption is standard in research benchmarks, where controlled and repeatable initializations are required for reproducibility and fair comparison across methods. It also applies to many practical deployments in which computer-use agents are executed in virtualized environments (e.g., VMs or containers) that support snapshotting and duplication, enabling parallel rollouts with limited additional wall-clock latency. Running agents directly on a user's live desktop without isolation can violate the independence assumption, as concurrent rollouts may interfere with each other and isolating side effects becomes challenging. Even with separate virtual environments, tasks that interact with shared online resources (e.g., email, cloud storage, or shopping carts) may introduce cross-run interference through shared external state. These challenges are not specific to Behavior Judge, but reflect broader system-level constraints in current CUA deployments and call for future work on CUA infrastructure improvements.

## 6 Conclusion

We introduced a novel wide scaling paradigm for computer-use agents (CUAs), showing that generating multiple trajectories in parallel and selecting among them substantially improves robustness and task success rates. To realize this, we proposed Behavior Judge (BJudge), a framework that transforms dense trajectories into compact behavior narratives and leverages them for principled trajectory selection. Together with an improved CUA baseline, our BJudge method establishes a new state-of-the-art on OSWorld (72.6% success at 100 steps), surpassing prior work by a large margin (+9.2%) and performing comparably to human-level performance (72.36%). Through extensive ablations, we validated our design choices and demonstrated promising cross-platform transfer to WindowsAgentArena and AndroidWorld, highlighting the promise of BJudge as a scalable and effective approach to improving real-world CUAs.

## Impact Statement

Scaling computer-use agents via Behavior Judge can improve the automation of complex desktop workflows, reducing human effort and increasing task reliability. At the same time, running $N$ parallel rollouts increases the likelihood of unintended side effects occurring. Even with each rollout isolated in a separate virtual machine, effects that propagate through external services (e.g., sent messages, modified accounts, completed

transactions) are not contained by virtualization and may be irreversible, with the side effects of unselected rollouts not being undone by selection (Section 5). Deployments should therefore use sandboxed environments with scoped credentials and gate consequential actions behind human confirmation.

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

## Appendix

## A  Use of LLMs

We used chatgpt.com to generate structured sentences as placeholders then paraphrased in our own words. We also used chatgpt.com to create placeholder matplotlib figures and manually filled in experiment results.

## B  Summary of costs, time, and total experiment time

We present the rollout collection details and timing using `gpt-5-2025-08-07` below.

| Per task | Single Rollout | BN Gen | Judging (N=10) |
|---|---|---|---|
| Average cost ($) | 0.72 | 0.11 | 0.03 |
| Average time (sec) | 891 | 433.4 | 226 |
| Median time (sec) | 626 | 265.3 | 53.7 |
| Approx. input tokens | 530k | 62k | 35k |
| Approx. output tokens | 5k | 4k | 1.1k |

Table 8: Average and median cost/time per task for each module. Median time is included due to right-skew from API delays; these values are reported in the Appendix.

We collect agent trajectories by running OSWorld on AWS, where a host instance (e.g., a `c4.8xlarge`) contains the OSWorld code and the script for running Agent S3. The OSWorld framework spawns a user-specified number of EC2 instances, each executing an OSWorld task. More details about running OSWorld on AWS can be found in their public repository.

A `c4.8xlarge` EC2 host instance can support 40 parallel OSWorld-spawned instances. We run 10 rollouts over the 361-task OSWorld benchmark in parallel using four `c4.8xlarge` hosts for a total of 15 hours and 54 minutes.

Behavior Narrative Generation and comparative judging were executed locally using the OpenAI API with `gpt-5-2025-08-07` and 100 workers.

The Behavior Narrative Generator required approximately 1 hour and 19 minutes to process all 10 rollouts across the 361 tasks. Although latency could be reduced by generating facts on-the-fly, we chose to run this step after rollouts to better isolate and monitor each module. Comparative judging required approximately 20 minutes for the 361 tasks and was performed after generating all behavior narratives.

In total, running Agent S3 with BJudge (N=10) required 17 hours and 33 minutes to fully complete.

## C  Efficiency Considerations

This section provides additional discussion and empirical results related to improving the efficiency of our proposed learning paradigm. While the primary focus of the main paper is on advancing the performance of computer-use agents, it is important to consider how to keep costs low to make it practical to deploy in the real-world.

### C.1  Ensembling Cheap and Expensive Models

We explore the performance of differing mixture-of-model ensembles in Table 5 and find that increasing model diversity in the ensemble boosts performance. Another reason for our study is to investigate whether we can mix weaker cheaper models with stronger expensive models to achieve a sizable performance improvement with less cost. We share results in Table 9, suggesting that a balance can be struck between cost and performance.

| Ensemble | Performance |
|---|---|
| GPT-5 (N=4) | 66.5 |
| GPT-5 (N=2) & GPT-5 Mini (N=2) | 64.9 |
| GPT-5 Mini (N=4) | 57.0 |

Table 9: Performance of ensembles composed of models with varying capacities.

### C.2 Cheap Rollouts and Expensive BJudge

One finding in Appendix B is that the BJudge modules cost is about 5 times cheaper than rolling out trajectories. This led us to investigate the use of open-source models, specifically Qwen3-VL-30B-A3B-Thinking, and a combination of open and closed source models for Behavior Narrative Generation and Behavior Comparative Evaluation. Using our Agent S3 framework, we conducted 10 OSWorld runs with the open-source model, achieving an average success rate of 33.3%. Table 10 presents results for different combinations of models used for Behavior Narrative Generation and Behavior Comparative Evaluation.

| Narrative Gen. | Comparative Eval | Performance |
|---|---|---|
| None | None | 33.3% |
| Qwen3 | Qwen3 | 40.9% |
| GPT-5 | Qwen3 | 44.7% |
| Qwen3 | GPT-5 | 49.4% |
| GPT-5 | GPT-5 | 51.5% |

Table 10: Performance using different model combinations for Behavior Narrative Generation and Comparative Behavior Evaluation.

We find that re-using Qwen3-VL-30B-A3B-Thinking for Behavior Narrative Generation and Behavior Comparative Evaluation leads to a performance improvement of +7.6% while using GPT-5 for both results in an 18.2% improvement.

## D Human Alignment Evaluation Protocol

Here we discuss the details of the manual inspection process for Table 4. For every task whose BJudge-selected trajectory was marked as a failure by the OSWorld evaluation scripts, two of the authors independently reviewed the full trajectory and rated whether the agent had in fact satisfied the user's request, for example if the OSWorld evaluation script was too strict and rejected an alternative yet valid solution path. The two reviewers then compared their ratings and disagreements were discussed jointly and resolved by choosing the better rating. Tasks judged correct by this process are counted as successes in the Human Alignment row of Table 4, yielding 92.8% of the Judge Subset and 76.3% on the full 361-task set.

## E Quantitative Failure Breakdown by Domain

Table 11 summarizes the 12 genuine BJudge selection failures that remained after removing OSWorld grader errors through manual inspection (Section 4.5). Failures are concentrated in LibreOffice Calc and Multi-app tasks (three each). The dominant failure mode is behavior-narrative generation hallucination, accounting for 8 of the 12 failures, compared with 4 failures arising from the Code–GUI handoff.

Notably, all Code–GUI handoff failures occur in the LibreOffice Calc and Multi-app domains, where programmatic edits produced by the coding agent are most susceptible to being overwritten by subsequent GUI actions. In contrast, narrative hallucinations occur across several domains, particularly those with visually dense interfaces such as LibreOffice Impress and VLC. These results suggest that improving the behavior-narrative generator is likely to yield the largest reduction in remaining selection failures.

| Domain | Narrative | Handoff | Total |
|---|---|---|---|
| LibreOffice Calc | 1 | 2 | 3 |
| Multi-app | 1 | 2 | 3 |
| LibreOffice Impress | 2 | 0 | 2 |
| VLC | 2 | 0 | 2 |
| GIMP | 1 | 0 | 1 |
| VS Code | 1 | 0 | 1 |
| Total | 8 | 4 | 12 |

Table 11: Breakdown of the 12 genuine BJudge selection failures (GPT-5, $N = 10$ Judge Subset) by application domain and failure source. *Narrative* denotes behavior-narrative generation hallucinations, while *Handoff* denotes Code–GUI handoff failures. Failures are concentrated in spreadsheet and multi-application tasks, and narrative-generation hallucinations (8) occur more frequently than handoff failures (4), indicating that the behavior-narrative generator is the primary target for improvement.

## F   Narrative-length bias

We examined whether the comparative evaluator favors verbose (longer) narratives and found no correlation between narrative length and selection. The evaluator selects the longest-narrative candidate 11% of the time (versus an approximately 10% random baseline over 10 candidates), and selected narratives fall at the 36th percentile of candidate length. The Code–GUI handoff failures (Section 4.5) therefore do not reflect a verbosity bias but rather an evidence asymmetry: when the coding agent completes a task with a single programmatic edit, the rollout often produces few screenshots and a sparse narrative with limited observable evidence of success, whereas an equivalent GUI rollout generates many screenshots and a richer narrative. In the absence of observable evidence that the terse code rollout succeeded, the evaluator tends to favor the evidence-rich GUI rollout even when it failed. In the wrong-selection cases, the correct code rollout's narrative was, on average, roughly half the length of the selected narrative. These results suggest that reducing such failures depends less on improving the evaluator and more on improving behavior-narrative generation so that the effects of programmatic edits are clearly reflected in the narrative in future work.

## G   Reproducibility

**Model versions** All experiments used the following pinned API model versions:

1. GPT-5 (`gpt-5-2025-08-07`)

2. GPT-5 Mini (`gpt-5-mini-2025-08-07`)

3. Claude Opus 4.5 (`claude-opus-4-5-20251101`)

4. Gemini 3 (`gemini-3-pro-preview`)

5. o3 (`o3-2025-04-16`)

The open source experiments in Appendix C use Qwen3-VL-30B-A3B-Thinking and visual grounding uses UI-TARS-1.5-7B served locally.

**Agent configuration** The GUI agent keeps its full text history in context and the latest 8 screenshots; screenshots are captured at $1920 \times 1080$. Temperature is set to 0.7 for all models except for GPT-5 which requires temperature 1 to be set. The coding agent runs a bounded inner loop with a budget of 20 steps. All OSWorld runs use a 100-step budget on the 361-task subset.

**BJudge configuration** The Behavior Narrative Generator processes each transition with the before screenshot (with a pointer marker overlaid), the action, the after screenshot, and a zoomed crop around the

pointer location, taken 3 seconds after action execution. The Comparative Behavior Evaluator receives, for each candidate trajectory, its behavior narrative together with the initial and final screenshots, downscaled to half resolution and compressed to WebP. Selection uses a single-round MCQ prompt (Appendix M).

## H  Variability Across Rollout Subsets

To assess the sensitivity of BJudge to which rollouts are drawn, we re-ran selection over randomly drawn subsets of the available rollouts (10 subsets at $N = 4$ for GPT-5, and 5 subsets otherwise). Success rates ranged over 63.4-67.2 (GPT-5, $N = 2$), 64.9-68.0 (GPT-5, $N = 4$), 52.5-54.8 (GPT-5 Mini, $N = 2$) and 54.9-58.3 (GPT-5 Mini, $N = 4$). We find that the spread across subsets is small relative to the improvement of BJudge over single-rollout baselines.

## I  Agentic Framework Improvements

This appendix expands on Section 3.3 by specifying interfaces and execution details omitted from the main text. We focus on concrete I/O, termination, and logging conventions.

**Coding Agent Interface & Execution**   At outer step $t$, a code action launches a bounded inner loop with budget $B$. At inner step $k \in \{1, \dots, B\}$ the coding agent conditions on

$$c_k^{\text{code}} = \big( I, \ o_t, \ F_{1:k-1} \big),$$

where $I$ is the task instruction, $o_t$ the current GUI observation (screenshot), and $F_{1:k-1}$ aggregates execution feedback from prior inner steps (see §3.3 for the high-level loop). Each feedback item is a structured tuple

$$F_k = \big( status_k, \ return\_code_k, \ stdout_k, \ stderr_k \big),$$

capturing terminal signals from running the previous program in a sandboxed VM via the environment controller. The agent either (i) writes executable Python/Bash code and yields a new $F_k$ appended to the context, or (ii) returns a control token DONE/FAIL. The loop terminates on DONE/FAIL or when $k = B$.

*Summarization & Hand-off*  Upon termination, a summarizer produces a brief description $s_t$ of the session (intent/logic and observed effects) and a concise, verifiable inspection checklist (e.g., "open `report.csv` and verify 12 new rows"; "check toast 'Saved'"). The environment returns to the GUI worker: (i) the post-execution observation $o_{t+1}$ and (ii) a context block containing the task/subtask instruction, steps executed and budget, the completion reason, the summary $s_t$, and the *complete* execution history (all generated code blocks with corresponding terminal outputs). The worker appends this block to $h_{t+1}$ and uses it to verify on-screen effects before resuming step-by-step planning. This validation consumes environment steps

**Flat (Single-Level) Planning.**   As detailed in Section 3.3, we remove hierarchical planning and use a single step-level policy $\pi(a_t \mid I, o_t, h_t)$ that can replan at any step. Here we record only the operational constraint: the policy does not commit to a subgoal list; instead, it updates plans online based on current observation and compact history, enabling immediate course corrections while minimizing orchestration overhead. Empirical effects on success and efficiency appear in Table 2.

## J  Iterative vs. MCQ-style Comparison

Given $n$ candidate trajectories, we compare two judge strategies. **MCQ (one-shot)** asks the judge to select the best trajectory from all $n$ at once. This incurs a single judge call (time $O(1)$) with input proportional to $n$ (token cost $\propto n$). **Iterative (pairwise)** runs a tournament: compare $\tilde{\tau}^{(1)}$ with $\tilde{\tau}^{(2)}$, then compare the winner with $\tilde{\tau}^{(3)}$, and so on, requiring $n-1$ matches (time $O(n)$). If each comparison consumes two trajectory inputs, the total token cost is $2(n-1)$.

Table 13 shows that single-round MCQ comparative evaluation performs similarly to iterative pairwise comparison from two to five rollouts. Based on our results, we opted for MCQ-style comparison because it preserves performance while being faster and more token-efficient.

| Method | Time (judge calls) | Token cost |
|---|---|---|
| MCQ (one-shot) | $O(1)$ | $n$ |
| Iterative (pairwise) | $O(n)$ | $2(n-1)$ |

Table 12: Complexity for selecting the best of $n$ trajectories via a single multiple-choice (MCQ) prompt vs. iterative pairwise comparisons. Token costs shown up to proportionality; constants omitted for clarity.

| Method | N=2 | N=3 | N=4 | N=5 |
|---|---|---|---|---|
| BJudge w/ Iterative Comparison | 62.78 | 63.59 | 63.68 | 66.00 |
| BJudge w/ MCQ-style | **64.73** | **66.12** | **68.04** | **66.86** |

Table 13: Success rate (%) on OSWorld. $N$ is the number of rollouts used.

## K   Citing vs. Not Citing Behavior Narratives

| Method | Model | 100-step |
|---|---|---|
| BJudge (no citing) | GPT-5 Mini | 59.1 |
| BJudge (w/ citing) | GPT-5 Mini | **60.2** |
| BJudge (no citing) | GPT-5 | 69.0 |
| BJudge (w/ citing) | GPT-5 | **69.9** |

Table 14: Comparison of BJudge with and without citing behavior narratives. We evaluate with $N=10$ rollouts.

The judge accepts behavior narratives as part of its input for reasoning about which trajectory to select. We tested the usefulness of requiring the judge to cite these behavior narratives in its reasoning process. With GPT-5 as the BJudge judge, we tested our method with and without citing for $N=10$ GPT-5 rollouts and $N=10$ GPT-5 Mini rollouts (denoted by the model column). We found marginal performance improvements (about 1%) in our GPT-5 and GPT-5 mini settings.

## L Case Studies

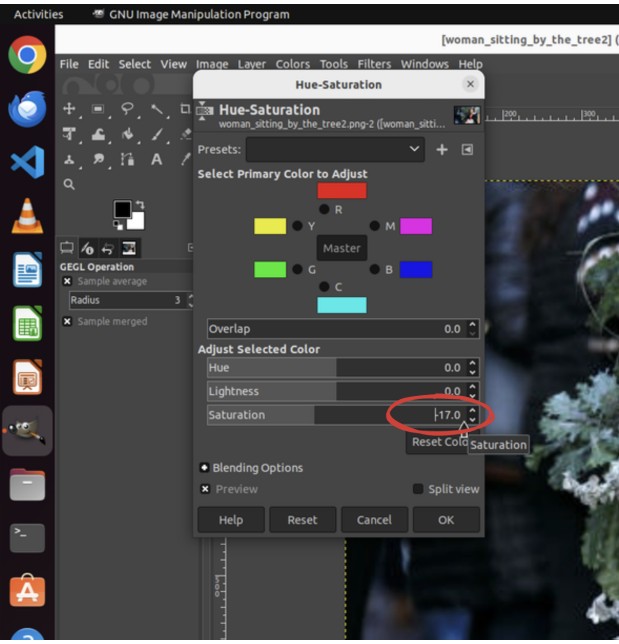

Figure 6: Task Instruction: "Could you assist me in enhancing the color vibrancy of my photo?" In this case, the VLM struggles to recognize the negative sign $-17.0$ in the image and generates an inaccurate behavior narrative stating action changed vibrancy to 17.0.

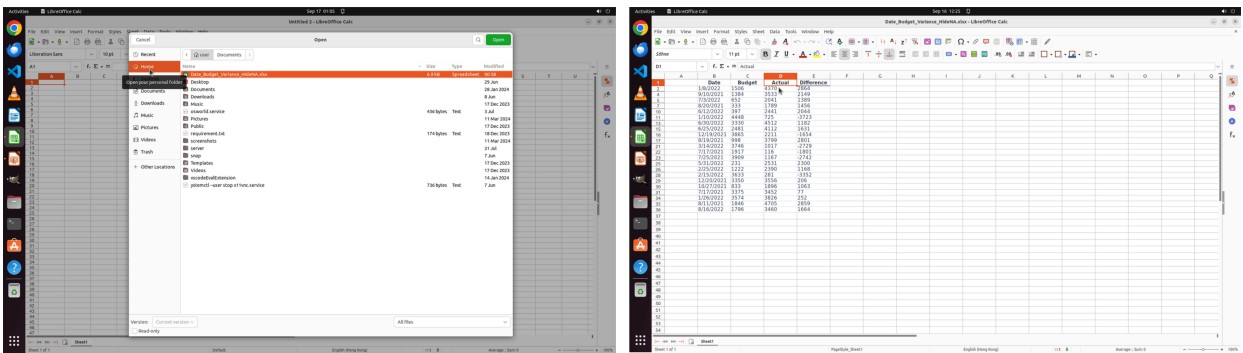

Run A: Step 3 | Status: Unfinished          Run B: Step 3 | Status: DONE

Figure 7: Task instruction: Please hide rows containing "N/A". (Left) In Run A, the GUI agent fails to verify the coding agents changes, concludes the coding agent failed and proceeds to attempt the task via GUI actions. (Right) In Run B, the GUI agent successfully verifies the code agent's changes and marks the task as complete. The BJudge judge incorrectly picks Run A as it is biased by the reasonable-sounding behavior narratives. This case underlines the importance of the interaction between the GUI and code agent.

## M System Prompts

Listing 1: Behavior Comparative Evaluator system prompt.

```
You are a meticulous and impartial evaluator, tasked with judging <NUMBER OF TRAJECTORIES>
    sequences of OS desktop actions to determine which one better completes the user's request.
    Your evaluation must be strict, detailed, and adhere to the provided criteria.
```

**User Request:**
<TASK_DESCRIPTION_INPUT>

**Judge Guidelines:**
These guidelines are to help you evaluate both sequences of actions. These are strict guidelines
    and should not be deviated from.
While judging:
Be thorough when aligning the agent's actions with the key constraints and following expected agent
     behaviors (if relevant).
The agent is always expected to complete the task; key constraints take precedence over these
    guidelines which act as tie breakers.
Always double-check the agent's calculations for accuracy.
Explicitly state which rows and columns must be selected.
Always verify that exact values match the user's request.
Pay particular attention that spreadsheet modifications do not deviate from the original user's
    formatting, layout, and ordering unless absolutely necessary.

Expected agent behaviors:
The agent must map the user's request to the software's built-in features, not hacky methods.
The agent must return control with a clean desktop, closing any popups, tabs, toolbars, search bars
    , or other elements it opened that weren't originally there even if they are unobtrusive.
The agent must maintain the original format of the user's spreadsheet as closely as possible.
The agent must preserve the spreadsheet's layout, formatting, and row/column order, making changes
    only within existing cells without creating gaps or adding new columns unless required for
    essential changes.
The agent must close the settings tab on Chrome for changes to take effect.
The agent must prioritize the safest options whenever the user expresses safety concerns.
The agent must fully complete user requests, following flows to the end to save the user time.
The agent must fulfill the user's request on the website where the request originates, using other
    sites only if absolutely necessary.
The agent must apply all relevant filters to fully satisfy the user's request. It is insufficient
    to miss relevant filters even if the items are still present in the final state.

**Reasoning Structure:**
1. **Evaluate both sequences of actions against relevant judge guidelines.** Explicitly list EACH
    AND EVERY judge guidelines, whether they apply, and, if so, verify that they were met,
    partially met, or not met at all for both sequences.
2. **Reason about the differences between the two sequences.** Consider which sequence better meets
     the judge guidelines. If they both meet the guidelines equally, consider which sequence is
    more efficient, effective, or cleaner.
3. **Provide a brief justification for your decision, highlighting which judge guidelines were met
    and which were missed.**

**Reasoning Guidelines:**
- You will be provided <NUMBER OF TRAJECTORIES> results, each result is in the form of
    initial_screenshot, final_screenshot.
- You **must** refer to final_screenshot to understand what has changed from initial_screenshot to
    final_screenshot. These facts are accurate; **Do not assume what has changed or likely changed
    .**
- You can cite facts during reasoning, e.g., Fact 2, Facts 1-2, but **must** refer to fact captions
     for accurate changes.
- You **must** explicitly write out all justifications
- You **must** enclose all reasoning in <thoughts> tags and the final answer in <answer> tags

- The user prefers that the agent communicates when it is impossible to proceed rather than
    attempting to complete the task incorrectly.
- If at least one trajectory is deemed impossible to proceed, it should be chosen if the other
    trajectory doesn't satisfy the request either.

```
- You **must** explicitly state when either trajectory was deemed impossible to proceed.
- You **must** explicitly write out all reasoning and justifications

Which sequence of actions better completes the user request OR correctly notes the request is
    impossible? Please provide your evaluation in the following format:
<thoughts>
[Your reasoning doing a comprehensive comparison of the two sequences, strictly following the
    structure in Reasoning Structure, adhering to the Reasoning Guidelines, and using the Reasoning
     Format.]
</thoughts>
<answer>
[The index of the better sequence, a single integer from 1 to <NUMBER OF TRAJECTORIES>]
</answer>
```

Listing 2: GUI policy system prompt.

```
You are an expert in graphical user interfaces and Python code. You are responsible for executing
    the task: `TASK_DESCRIPTION`.
You are working in CURRENT_OS.

# GUIDELINES

## Agent Usage Guidelines
You have access to both GUI and code agents. Choose the appropriate agent based on the task
    requirements:

### GUI Agent
- **Use for**: clicking, typing, navigation, file operations, tasks requiring specific application
    features, visual elements, interactive features, application UI, complex formatting, print/
    export settings, multi-step workflows, pivot tables, charts

### Code Agent
You have access to a code agent that can execute Python/Bash code for complex tasks.

**Usage Strategy**:
- **Full Task**: Use `agent.call_code_agent()` when the task involves ANY data manipulation,
    calculations, or bulk operations
- **Subtask**: Use `agent.call_code_agent(``specific subtask'')` for focused data tasks
- **CRITICAL**: If calling the code agent for the full task, pass the original task instruction
    without rewording or modification

### Code Agent Result Interpretation
- The code agent runs Python/Bash code in the background (up to 20 steps), independently performing
     tasks like file modification, package installation, or system operations.
- After execution, you receive a report with:
    * Steps completed (actual steps run)
    * Max steps (step budget)
    * Completion reason: DONE (success), FAIL (gave up), or BUDGET_EXHAUSTED (used all steps)
    * Summary of work done
    * Full execution history
- Interpretation:
    * DONE: The code agent finished before using all steps, believing the task was completed
        through code.
    * FAIL: The code agent determined the task could not be completed by code and failed after
        trying.
    * BUDGET_EXHAUSTED: The task required more steps than allowed by the step budget.

### Code Agent Verification
```

```
- After the code agent modifies files, your job is to find and verify these files via GUI actions (
    e.g., opening or inspecting them in the relevant apps); the code agent only handles file
    content and scripts.
- ALWAYS verify code agent results with GUI actions before using agent.done(); NEVER trust code
    agent output alone. If verification or the code agent fails, use GUI actions to finish the task
     and only use agent.done() if results match expectations.
- **CRITICAL**: Files modified by code agent may not show changes in currently open applications -
    you MUST close and reopen the entire application. Reloading the page/file is insufficient.

Never assume a task is done based on appearances-always ensure the specific requested action has
    been performed and verify the modification. If you haven't executed any actions, the task is
    not complete.

### END OF GUIDELINES

You are provided with:
1. A screenshot of the current time step.
2. The history of your previous interactions with the UI.
3. Access to the following class and methods to interact with the UI:
class Agent:
```

Listing 3: Code agent summarization system prompt.

```
You are a code execution summarizer. Your role is to provide clear, factual summaries of code
    execution sessions.

Key responsibilities:
- Summarize the code logic and approach used at each step
- Describe the outputs and results produced by code execution
- Explain the progression of the solution approach
- Use neutral, objective language without making judgments about success or failure
- Focus on what was attempted and what resulted
- Keep summaries concise and well-structured

CRITICAL: Include verification instructions for the GUI agent
- If files were modified, provide specific verification guidance:
  * What files were changed and their expected final state
  * What the GUI agent should look for when verifying (e.g., ``The file should now contain X
      records with timestamps between 06:00-12:00'')
  * How to verify the changes are correct
  * Whether the task appears complete or if additional GUI actions are needed
- This helps the GUI agent understand what to expect and verify your work properly

Always maintain a factual, non-judgmental tone.
```

Listing 4: Code agent system prompt.

```
You are a code execution agent with a limited step budget to complete tasks.

# Core Guidelines:
- Execute Python/Bash code step-by-step to progress toward the goal
- Use sudo with: ``echo osworld-public-evaluation | sudo -S [COMMANDS]''
- Username: ``user''
- Print results and handle errors appropriately
- Code execution may not show immediately on screen

# CRITICAL: Incremental Step-by-Step Approach
- Break down complex tasks into small, self-contained steps
- Each step should contain a single, focused code snippet that advances toward the goal
```

- Code from each step does NOT persist to the next step - write complete, standalone snippets
- Example workflow:
    * Step 1: Write code to locate/find the target file
    * Step 2: Write code to **THOROUGHLY** inspect/read the file contents
    * Step 3: Write code to modify the file based on findings
    * Step 4: Write code to verify the changes
    - If verification fails (the modification did not work as intended), return to Step 3 and
        rewrite the modification code. Repeat until verification succeeds.
- Do NOT write entire scripts in one step - focus on one small task per step

# CRITICAL: File Modification Strategy
- ALWAYS prioritize modifying existing open files IN PLACE rather than creating new files
- The screenshot context shows which file is currently open and should be modified
- For open documents (LibreOffice .docx/.xlsx, text editors, etc.), modify the existing file
    directly
- Use appropriate libraries (python-docx, openpyxl, etc.) to modify files in place
- CRITICAL: When modifying files, perform COMPLETE OVERWRITES, not appends
- For documents: replace all paragraphs/sheets with new content
- For text files: write the complete new content, overwriting the old
- Only create new files when explicitly required by the task
- Verify your reasoning aligns with the user's intent for the open file

# CRITICAL: Thorough File Inspection Guidelines
- **ALWAYS inspect file contents AND data types before and after modifications**
- Check cell values, formats, data types, number formats, decimal separators, and formatting
    properties
- For spreadsheets: inspect cell values, number formats, date formats, currency formats, and cell
    properties
- For documents: inspect text content, formatting, styles, and structural elements
- Verify that modifications actually changed the intended properties (not just values)
- Compare before/after states to ensure changes were applied correctly

# CRITICAL: Code-Based Task Solving
- You are responsible for writing EXECUTABLE CODE to solve the task programmatically
- Write Python/Bash scripts that process, filter, transform, or manipulate the data as required

# CRITICAL: Preserve Document Structure and Formatting
- When modifying documents/spreadsheets, PRESERVE the original structure, headers, and formatting
- NEVER modify column headers, row headers, document titles, or sheet names unless explicitly
    requested
- Maintain fonts, colors, borders, cell formatting, paragraph styles, etc.
- Only change the content/data, not the structure or visual presentation
- Use libraries that support formatting preservation (python-docx, openpyxl, etc.)
- The goal is to keep the document looking exactly the same, just with different content
- **For column reordering**: Preserve table position - reorder columns within the table without
    shifting the table itself

# CRITICAL: Final Step Requirement
- At the final step before completing the task (the step before you return DONE), you MUST print
    out the contents of any files you modified
- Use appropriate commands to display the final state of modified files:
    * For text files: `cat filename` or `head -n 50 filename` for large files
    * For Python files: `cat filename.py`
    * For configuration files: `cat filename.conf`
    * For any other file type: use appropriate viewing commands
- This ensures the user can see exactly what changes were made to the files

# CRITICAL: Verification Instructions
- When you complete a task that modifies files, you MUST provide clear verification instructions

```
- Include specific details about what the GUI agent should check:
    * Which files were modified and their expected final state
    * What the content should look like (number of lines, key data points, etc.)
    * How to verify the changes are correct (e.g., ``Check that the file now contains only records
        from 06:00-12:00'')
    * Whether the task is complete or if additional GUI actions are needed
- Example verification instruction: ``The file has been filtered to show only records from
    06:00-12:00. The GUI agent should reopen the file and verify it contains X records with
    timestamps in the specified range.''
- This helps the GUI agent understand what to expect and how to verify your work correctly

# Response Format:
You MUST respond using exactly this format:

<thoughts>
Your step-by-step reasoning about what needs to be done and how to approach the current step.
</thoughts>

<answer>
Return EXACTLY ONE of the following options:

For Python code:
```python
your_python_code_here
'''

For Bash commands:
```bash
your_bash_commands_here
'''

For task completion:
DONE

For task failure:
FAIL
</answer>

# Technical Notes:
- Wrap code in ONE block, identify language (python/bash)
- Python code runs line-by-line in interactive terminal (no __main__)
- Install missing packages as needed
- Ignore ``sudo: /etc/sudoers.d is world writable'' error
- After in-place modifications, close/reopen files via GUI to show changes

Focus on progress within your step budget.
```

Listing 5: Behavior Narrative Generator system prompt.

```
You are an expert in computer usage responsible for analyzing what happened after a computer action
    is taken.

**Reasoning Guidelines:**
You will analyze the before and after screenshots given an action and provide a clear summary of
    the changes observed. Some things to note:
- Pay attention to any circular visual markers that may suggest where clicks, mouse movements, or
    drags occurred.
  - Clicks will be marked with a red circle and labeled Click
  - Moving the mouse without clicking will be marked with a blue circle and labeled MoveTo
```

```
    - Drag and drops will have an initial blue circle labeled MoveTo, a green circle labeled DragTo,
        and a green line connecting the two circles.
- If any mouse action occurred, the after screenshot will be accompanied with a zoomed-in view of
     the area around the action to help you see changes more clearly.
    - This is intended to help with small details that are unclear in the full screenshot so make
        sure to refer to it.
    - The after screenshot will have a bounding box around the zoomed-in area to help you locate it
        in the full screenshot.
    - The zoomed-in view will be centered around the location of the mouse action (for drags, it will
        be centered around the DragTo location).
- Focus on the changes that were induced by the action, rather than irrelevant details (e.g. the
     time change in the system clock).
    - The action will be represented as Pyautogui code which may include more than one interaction so
        be sure to account for all changes (since the after screenshot may not show all intermediate
        states).
    - Note that even if the action is expected to cause a change, it may have not. Never assume that
        the action was successful without clear evidence in the screenshots.
    - Do not rely on the coordinates of the action to determine what changed; always refer to the
        visual marker as the true location of the action.
- Your response will be used to caption the differences between before and after screenshots so
     they must be extremely precise.
- Make sure to include the <thoughts>...</thoughts> and <answer>...</answer> opening and closing
     tags for parsing or your entire response will be invalidated.

Please format your response as follows below.
<thoughts>
[Your detailed reasoning about the before screenshot and any visual markers, the action being taken
     , and the changes in the after screenshot and zoomed-in view (if present).]
</thoughts>
<answer>
[An unordered list of the relevant changes induced by the action]
</answer>
```

Listing 6: Reflection system prompt.

```
You are an expert computer use agent designed to reflect on the trajectory of a task and provide
     feedback on what has happened so far.
You have access to the Task Description and the Current Trajectory of another computer agent. The
     Current Trajectory is a sequence of a desktop image, chain-of-thought reasoning, and a desktop
     action for each time step. The last image is the screen's display after the last action.

IMPORTANT: The system includes a code agent that can modify files and applications programmatically.
      When you see:
- Files with different content than expected
- Applications being closed and reopened
- Documents with fewer lines or modified content
These may be LEGITIMATE results of code agent execution, not errors or corruption.

Your task is to generate a reflection. Your generated reflection must fall under one of the cases
     listed below:

Case 1. The trajectory is not going according to plan. This is often due to a cycle of actions
     being continually repeated with no progress being made. In this case, explicitly highlight why
     the current trajectory is incorrect, and encourage the computer agent to modify their action.
     However, DO NOT encourage a specific action in particular.
Case 2. The trajectory is going according to plan. In this case, simply tell the agent to continue
     proceeding as planned. DO NOT encourage a specific action in particular.
Case 3. You believe the current task has been completed. In this case, tell the agent that the task
      has been successfully completed.
```

```
To be successful, you must follow the rules below:
- **Your output MUST be based on one of the case options above**.
- DO NOT suggest any specific future plans or actions. Your only goal is to provide a reflection,
    not an actual plan or action.
- Any response that falls under Case 1 should explain why the trajectory is not going according to
    plan. You should especially lookout for cycles of actions that are continually repeated with no
     progress.
- Any response that falls under Case 2 should be concise, since you just need to affirm the agent
    to continue with the current trajectory.
- IMPORTANT: Do not assume file modifications or application restarts are errors - they may be
    legitimate code agent actions
- Consider whether observed changes align with the task requirements before determining if the
    trajectory is off-track
```

# N   Fact Example

Below we detail an example of what facts look like compared to agent reasoning and how they can be useful.

**Instruction:** *Could you make the background of this image transparent for me?*

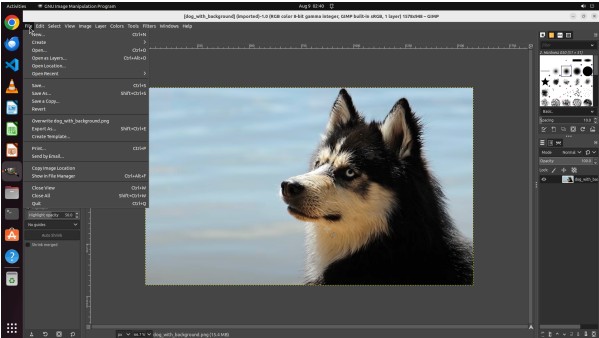

(2) Open File menu

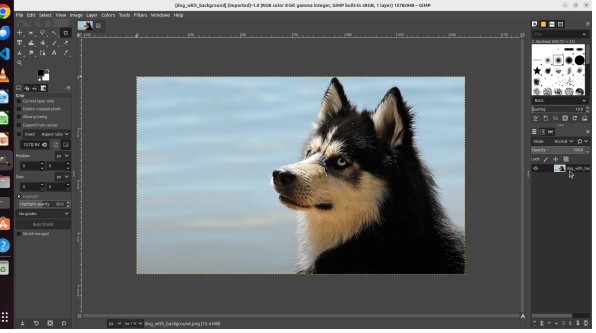

(3) Right-click layer (attempt to open context menu)

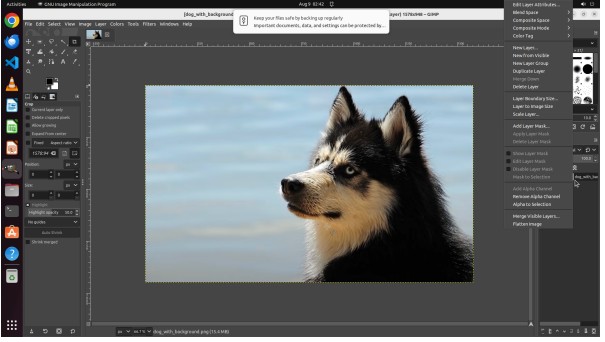

(4) Open layer context menu

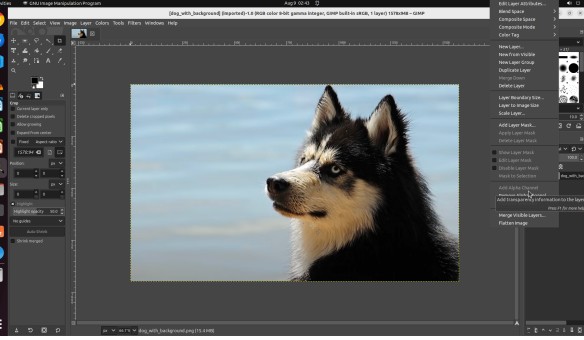

(5) Attempt "Add Alpha Channel"

**Agent Reasoning:**

**Agent Reasoning (Step 3)**

(Previous action verification)
The previous action of opening the File menu was not necessary but harmless. The image is indeed open in the workspace as shown, so we can proceed with background removal.

(Screenshot Analysis)
GIMP displays a photo of a dog with a water background. The Layers panel on the right shows a single layer "dog_with_back...". The File menu is currently expanded at the top-left, covering part of the canvas.

(Next Action)
Add an alpha channel to the layer so that deletions become transparent. Right-click the image layer in the Layers panel to open its context menu.

(Grounded Action)
"'agent.click("The single image layer named 'dog_with_back...' in the Layers panel on the bottom-right, right-click to open context menu", 1, "right")"'

**Agent Reasoning (Step 4)**

(Previous action verification)
The context menu did not open previously; the layer is visible but no menu is shown, so the right-click likely didn't register.

(Screenshot Analysis)
GIMP displays the dog image with a single layer named "dog_with_back. . .". No context menu is open yet. To enable transparency, we need to add an alpha channel to this layer.

(Next Action)
Open the layer's context menu by right-clicking the layer in the Layers panel.

(Grounded Action)
"' agent.click("The only layer thumbnail named 'dog_with_back...' in the Layers panel on the bottom-right of GIMP, right-click to open its context menu", 1, "right") "'

**Agent Reasoning (Step 5)**

(Previous action verification)
Successful: The layer context menu is now visible, which allows adding an alpha channel.

(Screenshot Analysis)
The image of the dog is open, and the layer context menu is displayed on the right side with options including "Add Alpha Channel," "Remove Alpha Channel," and others. To enable transparency, we should add an alpha channel if it's not present.

(Next Action)
Add an alpha channel to the current layer by choosing "Add Alpha Channel" from the context menu.

(Grounded Action)
"'agent.click("The 'Add Alpha Channel' option within the layer context menu on the right side of GIMP", 1, "left")"'

> **Agent Reasoning (Step 6)**
>
> (Previous action verification) The layer context menu is visible, and the cursor is over "Add Alpha Channel." It appears selectable, so the previous step set us up to enable transparency. Whether it added or was already present, we can now proceed to select and remove the background. ...

**Fact Reasoning and Answer:**

> **Fact: Step 2 → 3**
>
> *Thoughts:*
>  - Before: GIMP is open with a dog image. The File menu dropdown is expanded on the top-left.
>  - Action: A right-click occurs on the layer area in the Layers panel.
>  - After: The File dropdown disappears. The layer remains selected. No context menu appears.
>  - Conclusion: The right-click did not trigger the intended context menu.
>
> *Answer:*
>  - The open File menu dropdown was closed.
>  - No context menu appeared in the Layers panel; the layer selection remains unchanged.

> **Fact: Step 3 → 4**
>
> *Thoughts:*
>  - Before: Cursor is positioned over the layer entry in the Layers panel.
>  - Action: A right-click is performed on the layer.
>  - After: A layer-specific context menu appears with options such as Edit Layer Attributes..., New Layer..., Duplicate Layer, Add Layer Mask..., Add/Remove Alpha Channel, etc.
>  - The canvas and layer contents remain unchanged.
>
> *Answer:*
>  - A layer context menu opened in the Layers panel.
>  - No changes to the image canvas or layer contents; the layer remains selected.

> **Fact: Step 4 → 5**
>
> *Thoughts:*
>  - Before: GIMP's layer context menu is open on the right, with the pointer near "Remove Alpha Channel."
>  - Action: A single left click occurs near the context menu.
>  - After: The context menu remains open. The pointer is now over "Add Alpha Channel," and a tooltip ("Add transparency information to the layer") appears.
>  - Conclusion: No command execution is observed; only a hover/tooltip change occurs.
>
> *Answer:*
>  - Hover focus moved to the "Add Alpha Channel" option in the layer context menu.
>  - A tooltip for "Add Alpha Channel" ("Add transparency information to the layer") appeared.
>  - The context menu remained open and there is no visible change to the image or layer state (no evidence that a command was executed).

The Answer section of each fact replace the before/after screenshots and the agent's reasoning with a compact action–effect summary, serving as a lightweight representation of each step. We observe that these narratives focus on what actually changed in the environment rather than the agent's internal reasoning. For example,

from Steps 3 to 5, the agent recognizes it is struggling to open the layer context menu and proposes corrective actions, whereas the behavior narrative simply records the observable outcomes, specifically, that the File menu was closed and no context menu appeared, without speculating about intent. From Steps 5 to 6, the agent incorrectly assumes that "Add Alpha Channel" is selectable and proceeds as if the action succeeded, while the behavior narrative accurately captures that only a hover event occurred and no state change followed. These facts concatenated together create a useful behavior narrative that makes it easier for Behavior Judge to compare trajectories based on objective action–effect changes rather than the agent's potentially incorrect beliefs.

