# OpenReview forum: "Scaling Agents for Computer Use"
_TMLR — Decision pending for TMLR_

### Review · Reviewer_U797 · 2026-05-08

**Summary Of Contributions:**

This paper studies test-time scaling for computer-use agents (CUAs) on long-horizon desktop and mobile tasks. Unlike traditional method, this paper introduces the wide scaling paradigm. Then it proposes Behavior Judge, enabling the evaluation of multiple rollouts. Empirically, this work can improve the original SOTA to 72.6 on OSWorld which also surpasses human performance. Additional ablations are used to validate the design choices.

Key strengths:
1. the clear motivation for trajectory-level scaling, the intuitive and practical behavior-narrative representation
2. strong benchmark performance that outperforms existing SOTA and human performance

Key weaknesses: I am not an expert in this field so I am afraid that I cannot identify very critical weaknesses of this work. One minor comment is that the  BJudge Accuracy in Table 4 is evaluated over 10 rollouts and it might be better if the author could include the std. or other stats in that table. Also, "We introduce wide scaling paradigm" sounds like introducing a new technique but it seems to be another variant of best-of-N or something-of-N.

**Audience:**

Yes

**Audience Explanation:**

Yes. The paper is relevant to several parts of the TMLR audience: researchers working on multimodal agents, GUI agents, test-time scaling, agent evaluation, long-horizon decision-making, and practical deployment of LLM/VLM-based systems.

**Broader Impact Concerns:**

Yes this paper has included a brief Impact Statement and I think it has reflected the scoop and the impact of this work.

**Claims And Evidence:**

Yes

**Claims Explanation:**

Overall, the central claims are supported by the experiments. The paper provides evidence that selecting among multiple CUA rollouts using behavior narratives improves over single-rollout execution, over an adapted WebJudge-style independent scoring baseline, and over several alternative trajectory representations.

Therefore, I am convinced that evidence supports the main technical direction and empirical effectiveness.

**Requested Changes:**

One minor comment is that the  BJudge Accuracy in Table 4 is evaluated over 10 rollouts and it might be better if the author could include the std. or other stats in that table.

 Also, "We introduce wide scaling paradigm" sounds like introducing a new technique but it seems to be another variant of best-of-N or something-of-N. --- So, I suggest the author could explicitly differentiate this method with other related literature.

---

> ### Author Response · Authors · 2026-06-15
>
> We thank the reviewer for their supportive review, address both suggestions below and have updated the paper to reflect the changes.
>
> > One minor comment is that the BJudge Accuracy in Table 4 is evaluated over 10 rollouts and it might be better if the author could include the std. or other stats in that table.
>
> We have added statistics to Table 4: the accuracies now carry bootstrap 95% confidence intervals computed over tasks (78.4% [71.7, 84.7] on the Judge Subset and 69.9% [65.0, 74.5] on the full set).
>
> > Also, "We introduce wide scaling paradigm" sounds like introducing a new technique but it seems to be another variant of best-of-N or something-of-N. --- So, I suggest the author could explicitly differentiate this method with other related literature.
>
> We have made the relationship to best-of-N explicit. Section 2.2 now notes that wide scaling can be viewed as a trajectory-level analogue of best-of-N sampling, but differs from standard best-of-N in that our candidates are full long-horizon trajectories drawn from multiple agents rather than i.i.d. samples from a single model; the following paragraph further emphasizes that the core difficulty is evaluating these trajectories rather than generating them. We retain the term "wide scaling" as, to our knowledge, this is the first work to study trajectory-level scaling of this kind for computer-use agents. We describe and differentiate against the closest prior work, GTA-1 [1], under ‘step-wise BoN’ which is the closest CUA approach to best-of-N.
>
> [1] GTA1: GUI Test-time Scaling Agent. Yang et al. 2025

---

### Review · Reviewer_8gvv · 2026-05-16

**Summary Of Contributions:**

This paper studies test-time scaling for computer-use agents (CUAs), focusing on the problem that single-rollout execution is brittle for long-horizon desktop tasks. The main idea is “wide scaling”: instead of committing to one rollout, the system generates multiple full trajectories from one or more base agents and then selects the best trajectory. To make trajectory-level selection feasible, the paper proposes Behavior Judge (BJudge), which first converts raw screenshots and actions into compact behavior narratives and then uses a comparative VLM-based evaluator to choose the best candidate trajectory.

The paper also introduces Agent S3, an improved CUA baseline built on Agent S2, with a flatter policy and an integrated coding agent for programmatic edits. Empirically, the method achieves strong results on OSWorld, reporting 72.6% success rate with GPT-5 + Claude Opus 4.5 rollouts, exceeding the previous reported SoTA and slightly surpassing the human-level number reported by OSWorld. The paper further provides ablations on behavior narratives, comparisons with WebJudge-style selection, resource allocation studies, mixture-of-model experiments, failure analysis, and generalization experiments on WindowsAgentArena and AndroidWorld.

The main strengths are:

1. The problem is timely and important, since reliable long-horizon computer-use agents remain difficult.
2. The proposed behavior narrative representation is simple, intuitive, and empirically useful.
3. The experimental results on OSWorld are strong, and the paper includes several useful ablations and cost/runtime details.
4. The paper explicitly analyzes failure modes such as narrative hallucination and code-GUI handoff failures.

**Additional Comments:**

Overall, I find the paper timely, practically relevant, and empirically strong. The behavior narrative representation is a clean and useful idea, and the experimental results suggest that trajectory-level selection is a promising direction for improving long-horizon CUAs.

My main concern is not whether the method works, but whether the paper currently isolates and states its contribution precisely enough. The strongest contribution is not simply “scaling agents,” but showing that full-trajectory scaling becomes effective when dense multimodal traces are converted into action-effect behavior narratives and judged comparatively. If the authors sharpen this framing, add stronger statistical/reproducibility evidence, and qualify the human-level and generalization claims, the paper would be a solid contribution to the CUA and test-time scaling literature.

**Audience:**

Yes

**Audience Explanation:**

This paper should be of interest to TMLR readers working on LLM/VLM agents, test-time scaling, computer-use agents, agent evaluation, and long-horizon decision-making. The core observation is useful: for long-horizon computer-use tasks, scaling over full trajectories can be more effective than only improving local step-level decisions, provided that there is a reliable trajectory-level selection mechanism.

The behavior narrative idea is also broadly relevant beyond OSWorld. It suggests a practical intermediate representation for judging long, multimodal agent traces, which may be useful for web agents, mobile agents, desktop automation, debugging, evaluation, and agent safety auditing. The empirical gains are large enough to be informative even if some claims require qualification.

The work is less theoretically deep than some TMLR submissions, but it is practically important and well aligned with current interest in agent reliability and test-time compute scaling.

**Broader Impact Concerns:**

This work makes computer-use agents more capable, which is useful for automation but also raises safety concerns. A CUA can modify files, send messages, interact with websites, or affect user accounts. With multiple rollouts, these risks may increase if the rollouts are not properly sandboxed.

**Claims And Evidence:**

Yes

**Claims Explanation:**

The paper provides convincing empirical evidence for the core claim that selecting among multiple rollouts can improve CUA performance when paired with a structured trajectory representation. The OSWorld results are strong: Agent S3 with BJudge substantially improves over Agent S3 alone, and the paper reports improvements for both GPT-5 and GPT-5 Mini settings. The comparison against WebJudge under equal rollout budgets, the representation ablation showing behavior narratives outperform screenshot-only, trajectory summary, and naive captioning baselines, and the resource-allocation study all support the practical usefulness of BJudge.

However, some claims should be qualified. First, the claim of surpassing human-level performance is not fully convincing as currently phrased. The reported margin over human performance is very small, and the paper itself acknowledges that OSWorld evaluation scripts can be imperfect. The authors should avoid presenting this as a decisive human-level result unless they provide statistical significance tests and a clearer human adjudication protocol.

Second, the evidence does not fully disentangle the benefit of BJudge from the benefit of Agent S3. Since Agent S3 itself is a strong new baseline with major improvements over Agent S2, the paper should more directly show that BJudge improves different base agents, not only the authors’ improved framework. Applying BJudge to Agent S2, another public CUA, or fixed rollout logs from prior systems would make the central claim more convincing.

Third, the generalization claim is promising but somewhat limited. The results on WindowsAgentArena and AndroidWorld show positive gains, but the experiments are smaller and use different settings, such as disabling the coding agent on AndroidWorld. The paper should therefore describe these results as evidence of promising cross-platform transfer rather than strong generalization in a broad sense.

Overall, the main empirical claims are supported, but several headline claims should be softened or backed with stronger statistical and reproducibility evidence.

**Requested Changes:**

### 1. Disentangle the contribution of BJudge from Agent S3

The paper introduces both BJudge and Agent S3, and Agent S3 itself provides a large performance boost. To make the contribution of BJudge clearer, the authors should evaluate BJudge on at least one additional base agent, such as Agent S2 or another public CUA framework. Alternatively, they could provide experiments using fixed rollout logs from an existing baseline system.

This would help show that BJudge is a general trajectory-selection method, rather than a method that mainly works together with the newly introduced Agent S3 framework.

### 2. Add statistical uncertainty and significance testing

The paper reports several close comparisons, especially the claim that the method surpasses human performance on OSWorld. The authors should report confidence intervals or bootstrap estimates over tasks, paired significance tests where applicable, and standard deviations across multiple rollout sets or random seeds.

This is particularly important for the comparison between the reported 72.6% success rate and the 72.36% human-level performance. Without statistical evidence, this difference is too small to strongly support the claim of surpassing human performance.

### 3. Qualify the “human-level” and “new SoTA” claims

The paper should avoid overstating the human-level claim unless the authors can show that the difference is statistically meaningful and robust to imperfections in the benchmark evaluator.

A more careful statement would be that BJudge reaches approximately human-level performance under the OSWorld automatic evaluation protocol.

### 4. Improve reproducibility

The paper relies heavily on proprietary models and API behavior. The authors should provide model versions, rollout sampling settings, cost details.

### 5. Analyze BJudge failure modes more systematically

The paper identifies narrative hallucination and code-GUI handoff failures, which are important failure modes. However, the analysis is still somewhat qualitative.

The authors should provide a more quantitative breakdown by task type, application domain, and failure source. It would also be helpful to report whether failures mainly come from the behavior narrative generator or from the comparative evaluator.

---

> ### Author Response · Authors · 2026-06-15
>
> We thank the reviewer for their thorough and constructive review, and for recognizing the timeliness of the problem, the behavior narrative representation, and the strength of the empirical results. We address each requested change below and have updated the paper to reflect the changes.
>
>
> > 1. Disentangle the contribution of BJudge from Agent S3 — evaluate BJudge on at least one additional base agent.
>
> For this study we generated a fresh set of 10 independent rollouts of Agent S2 on the OSWorld test set, and applied the original BJudge pipeline (GPT-5 behavior-narrative generation followed by single-round MCQ selection) to them. Across this rollout set (evaluated on the 361 tasks completed by all 10 rollouts), BJudge selects substantially better trajectories than the average rollout:
>
> | Method | Success rate |
> | --- | --- |
> | Mean single rollout (Agent S2, N=10) | 44.3 ± 0.6 |
> | Agent S2 + BJudge (N=10) | 56.5 [51.3, 61.6] |
> | Oracle selection (N=10, upper bound) | 66.6 [61.6, 71.3] |
>
> (Following Table 1's conventions: ± denotes the standard error of the mean over the 10 runs; brackets denote bootstrap 95% CIs over the tasks.)
>
> BJudge improves over the mean single rollout by +12.2 points, recovering 78% of the gain attainable by an oracle selector over the same rollouts. We have added the first two rows to the main results table (Table 1). We have noted that, unlike the other rows, these rollouts were generated using Azure GPT-5 access (due to restrictions), whose content-safety filtering intermittently blocks otherwise-valid agent actions and tends to yield somewhat lower absolute success than the direct `gpt-5-2025-08-07` access used everywhere else; the BJudge improvement over the matched single-rollout baseline is unaffected by this.
>
> > 2. Add statistical uncertainty and significance testing.
>
> We have added comprehensive statistical reporting (conventions stated in Section 4.1, Implementation Details):
>
> - Confidence intervals: all headline results carry bootstrap 95% CIs over the 361 tasks (10,000 resamples), reported in Tables 1 and 4 (e.g., best result 72.6 [68.0, 77.1]).
> - Paired significance tests: the key improvement claims now use paired bootstrap over per-task score differences. BJudge vs. Agent S3: +7.3% (95% CI [4.6, 10.0]) for GPT-5 and +10.4% (paired 95% CI [7.3, 14.1]) for GPT-5 Mini.
> - Uncertainty across runs: baselines are reported as mean ± standard error over 10 independent runs (62.6 ± 0.5 for GPT-5; 49.8 ± 0.7 for GPT-5 Mini).
> - Std across rollout sets: a new appendix ("Variability Across Rollout Subsets") reports BJudge performance over 5–10 randomly drawn rollout subsets at N=2 and N=4 for both models; because subsets necessarily overlap, we report observed ranges (e.g., GPT-5 N=4: 64.9–68.0) rather than standard errors.
>
> Regarding the 72.6% vs. 72.36% comparison specifically: per-task human outcomes are not available, so a paired test against the human baseline is not possible; we instead report our CI ([68.0, 77.1]), note that it contains the human-level score, and have removed all claims of surpassing human performance (see next point).
>
> > 3. Qualify the "human-level" and "new SoTA" claims.
>
> The paper now states that BJudge performs comparably to human-level performance (Section 4.2), with the abstract, introduction, conclusion, and Figure 1 caption revised accordingly; we no longer claim to surpass human performance. The generalization claims are likewise qualified with the WindowsAgentArena and AndroidWorld results now described as promising cross-platform transfer rather than strong generalization (abstract, introduction, Section 4.8, and conclusion).
>
> > 4. Improve reproducibility.
>
> We have added a "Reproducibility" appendix consolidating: pinned API model versions for every model used (e.g., gpt-5-2025-08-07, gpt-5-mini-2025-08-07, claude-opus-4-5-20251101, gemini-3-pro-preview), agent configuration (full text history with the latest 8 screenshots at 1920×1080, sampling temperature of 0.7 (except for GPT-5 with requires a temperature of 1.0) and the coding-agent step budget), and the BJudge pipeline configuration (visual augmentations for narrative generation, half-resolution WebP screenshots and single-round MCQ prompting for judging). Per-module cost and token details are in Appendix B.

---

> > ### Author Response · Authors · 2026-06-15
> >
> > > 5. Analyze BJudge failure modes more systematically.
> >
> > We have strengthened the failure analysis (Section 4.5) by making it quantitative and adding a breakdown table in the appendix. Across the 159 Judge Subset tasks where rollout outcomes differ, BJudge achieves 78.4% of the score attainable against the OSWorld scripts. Manual inspection shows that 23 flagged tasks are grader errors: the selected trajectory successfully completes the task through an alternative path that the script does not recognize. Accounting for these cases raises accuracy to 92.8% and leaves 12 genuine selection failures.
> >
> > We further categorize these 12 failures by source and application domain. Eight arise from behavior-narrative generation hallucinations, while four result from Code--GUI handoff failures. Failures are concentrated in LibreOffice Calc and Multi-app tasks (three each). All handoff failures occur in these two domains, whereas narrative hallucinations also appear in visually dense applications such as LibreOffice Impress and VLC.
> >
> > These results suggest that the primary limitation lies in the behavior-narrative generation stage rather than in the evaluator's selection logic. Eight failures are direct narrative-generation hallucinations. The remaining four occur at the generation-selection interface, where an unsuccessful rollout produced a more compelling narrative than a correct one-step rollout. Improving the fidelity of rollout narratives to actual task outcomes therefore appears to be the most promising direction for reducing the remaining errors.

---

### Review · Reviewer_9XRx · 2026-06-01

**Summary Of Contributions:**

The paper introduces a "wide scaling" paradigm for computer-use agents, generating multiple parallel trajectories and selecting the optimal outcome using a proposed "Behavior Judge" (BJudge). The paper also introduces an improved baseline framework, Agent S3, which drops hierarchical planning and natively integrates a coding agent.

Key strength in my opinion is the proposed shift from step-by-step evaluation to trajectory-level evaluation for computer-use agents. The "behavior narrative" idea of compressing each trajectory into action-effect facts before comparison is well-motivated, and the comparative evaluation result in Figure 4 is solid.

Main weakness lies in the paper's framing and evaluation of key metrics. The claim of surpassing human-level performance is based on 0.24% margin over baseline on a 361-task evaluation (which translates to roughly 1 task), and a difference this small should be within the range of random variation. The paper also reports no confidence intervals or statistical test for this comparison.

**Audience:**

Yes

**Audience Explanation:**

Researchers working on agentic software engineering, agent evaluation and GUI agents will find this topic very relevant.

**Broader Impact Concerns:**

The current impact statement in the paper is brief and insufficient. Rollouts that interact with external services can cause real-world consequences even across separate VMs. The authors should explicitly address the security, ethical, and operational risks of scaling autonomous code execution in real-world environments, even when virtualization is used.

**Claims And Evidence:**

Yes

**Claims Explanation:**

The core contributions, including comparative trajectory evaluation and behavior narratives, are supported reasonably well. While the methodological innovation and the empirical ablations are clear, some key success metrics are not soundly supported or framed.

1. The most claimed improvement in the paper is 72.6% against a human baseline of 72.36%, which is roughly one task out of 361, and a difference this small seems to be within the range of random variation I would expect at this sample size. The paper reports no confidence intervals or statistical test for this comparison.

2. In Section 4.5 a 92.8% human alignment figure is reported, but the manual inspection process is not described in enough detail for the readers to assess its reliability.

**Requested Changes:**

1. The 0.24% margin over human requires either adding a statistical test with confidence intervals, or reframing of the claimed "surpassing human performance."

2. (Minor) In Appendix B, it is revealed that generating 10 rollouts costs $7.20 per task. However, the main text does not integrate this economic overhead into the primary performance analysis. Should address and discuss this in the main text.

3. (Minor) Add some description in Section 4.5 for the manual inspection process and the 92.8% human alignment.

---

> ### Author Response · Authors · 2026-06-15
>
> We thank the reviewer for their careful reading and for recognizing the value of trajectory-level evaluation and the behavior narrative representation. We address your requested changes below and have updated the paper to reflect the changes.
>
> > The 0.24% margin over human requires either adding a statistical test with confidence intervals, or reframing of the claimed "surpassing human performance."
>
> We have done both. The paper no longer claims to surpass human performance: the abstract, introduction, Section 4.2, conclusion, and Figure 1 caption now state that our best result performs comparably to human-level performance (72.36%).
>
> We have also added statistical reporting throughout (conventions described at the end of Section 4.1, Implementation Details). All headline results now carry nonparametric bootstrap 95% confidence intervals computed over the 361 tasks (10,000 resamples); our best result is 72.6% with a 95% CI of [68.0, 77.1]. Baseline (non-scaling) results are reported as mean ± standard error over 10 independent runs (e.g., Agent S3 with GPT-5: 62.6 ± 0.5), and the key improvement claims are supported by paired bootstrap CIs over per-task score differences. BJudge improves over the Agent S3 baseline by +7.3% (95% CI [4.6, 10.0]) with GPT-5 and +10.4% (paired 95% CI [7.3, 14.1]) with GPT-5 Mini.
>
>
> > (Minor) In Appendix B, it is revealed that generating 10 rollouts costs \$7.20 per task. However, the main text does not integrate this economic overhead into the primary performance analysis.
>
> Section 4.2 now integrates the economic overhead into the main results that generating 10 rollouts averages \$$7.20 per task, while BJudge itself adds only ~\$0.14 per task for narrative generation and judging, with pointers to the performance-compute trade-off analysis (Section 4.6) and cheaper configurations (Appendix C). We additionally extended the Appendix B cost table with approximate per-module token counts (single rollout ≈530k input / 5k output tokens; narrative generation ≈62k / 4k; judging at N=10 ≈35k / 1.1k), reconstructed from the logged trajectories.
>
> > (Minor) Add some description in Section 4.5 for the manual inspection process and the 92.8% human alignment.
>
> We have added a new appendix, "Human Alignment Evaluation Protocol," referenced from Section 4.5. For every task whose selected trajectory was marked as a failure by the OSWorld evaluation scripts, two of the authors independently reviewed the full trajectory (instruction, screenshots, and actions) and rated whether the agent had in fact satisfied the request (e.g., via a valid alternative solution path not covered by the script's predefined checks); disagreements were discussed jointly and resolved by choosing the better-supported rating.
>
> > The current impact statement in the paper is brief and insufficient. Rollouts that interact with external services can cause real-world consequences even across separate VMs.
>
> We have expanded the Impact Statement to address exactly this concern. It now discusses how scaling to parallel rollouts increases the likelihood of unintended side effects; that effects propagating through external services are not contained by virtualization and may be irreversible, with the side effects of unselected rollouts not undone by selection; and concrete operational safeguards (sandboxed environments with scoped credentials and human confirmation for consequential actions).

---

### Review · Reviewer_uPri · 2026-06-15

**Summary Of Contributions:**

This paper introduces Behavior Judge (BJudge), a test-time "wide scaling" method for computer-use agents (CUA): each rollout is compressed into a behavior narrative, and a single-round MCQ evaluator selects the best trajectory by comparing narratives. Paired with a stronger base agent, it reports SoTA on OSWorld and cross-platform gains.

**Audience:**

Yes

**Audience Explanation:**

The overall topic of scaling CUA on long-horizon tasks is a trending topic and the behavior-narrative representation may be reusable for judging long multimodal traces beyond OSWorld.

**Broader Impact Concerns:**

The current two-sentence statement is not sufficient. For example, the risks of scaling autonomous multi-rollout execution against real services should be addressed explicitly.

**Claims And Evidence:**

Yes

**Claims Explanation:**

- The core claim of the paper *partially* holds: the narrative-based selection method beats single-rollout and WebJudge-style scoring under equal rollout budget.
- However, there are a few points that are not supported by the current experiments/results:
  - "Surpasses human" is an overclaim: 72.6 vs 72.36 is ~1 task/361, no CIs or significance, and in section 4.5, it calls the OSWorld evaluator imperfect.
  - BJudge's claims are framework-level (In Section 2.1, the first to scale over "multiple base models and policies"; In Section 3.2 formalizes it over arbitrary base policies), but every BJudge result uses Agent S3 rollouts only. Table 5 varies the LLM, not the base agent framework.
  - The system prompt in Appendix H contradicts the method described in the main body: it tells the judge to use "all screenshots in chronological order" and cite screenshots in pairwise form; However, Sections 3.1/3.2 claim a narrative-based, fact-citing, MCQ-over-N judge. This may undermine the results in Table 3 as well, since the success rates there are very close.

**Requested Changes:**

-  Fix the "surpasses human" claim: add CIs/bootstrap and significance tests, or restate as "approximately human-level under the OSWorld evaluator."
- Validate generality beyond Agent S3: apply BJudge unchanged to a different base agent (e.g., Agent S2 or a public CUA) or to fixed rollout logs from an existing system.
- Align the system prompt with the method itself, and confirm the version that produces the results in Table 3
- Quantify and address the judge's bias towards verbose narratives: In section 4.5 (Figure 7), it shows that BJudge picks a failed GUI-heavy rollout over a successful code-only one because the former produces "richer" narratives.

---

> ### Author Response · Authors · 2026-06-15
>
> We thank the reviewer for the careful reading and for recognizing the relevance of trajectory-level scaling and the reusability of the behavior-narrative representation. We address each requested change below and have updated the paper to reflect the changes.
>
> > Fix the "surpasses human" claim: add CIs/bootstrap and significance tests, or restate as "approximately human-level under the OSWorld evaluator."
>
> The paper no longer claims to surpass human performance; the abstract, introduction, Section 4.2, conclusion, and Figure 1 caption now state that our best result is comparably to human-level under the OSWorld evaluator (72.6% vs. 72.36%). We have also added bootstrap 95% confidence intervals over the 361 tasks (10,000 resamples) to all headline results (e.g., best result 72.6 [68.0, 77.1]), and report baselines as mean ± standard error over 10 runs (e.g., Agent S3 with GPT-5: 62.6 ± 0.5). Because per-task human outcomes are not available, a paired test against the human number is not possible; we instead note that our CI contains the human-level score and have removed the comparative claim.
>
> > Validate generality beyond Agent S3: apply BJudge unchanged to a different base agent (e.g., Agent S2 or a public CUA) or to fixed rollout logs.
>
> We generated 10 independent rollouts using Agent S2 on OSWorld and applied the BJudge pipeline unchanged. Over the 361 tasks completed by all 10 rollouts, BJudge improves the mean single rollout from 44.3% to 56.5% (+12.2 points, recovering 78% of the oracle selection gain). The first two rows are noted in Table 1. Applied with no modification to a different base agent, BJudge delivers a substantial improvement at no additional rollout cost, supporting the framework-level claim.
>
> | Method | Success rate |
> | --- | --- |
> | Mean single rollout (Agent S2, N=10) | 44.3 ± 0.6 |
> | Agent S2 + BJudge (N=10) | 56.5 [51.3, 61.6] |
> | Oracle selection (N=10, upper bound) | 66.6 [61.6, 71.3] |
>
> > Align the system prompt with the method itself, and confirm the version that produces the results in Table 3.
>
> We thank the reviewer for catching this. The listing previously shown in Appendix H was an outdated version of the evaluator prompt. The evaluator used for all narrative-based results, including Table 3, operates over behavior narratives and cites facts, consistent with Sections 3.1–3.2, and does not consume raw chronological screenshot sequences. We have replaced the Appendix H listing with the exact prompt used to produce the results and corrected the residual pairwise ("two sequences") wording to reflect the single-round MCQ-over-N formulation.
>
> > Quantify and address the judge's bias towards verbose narratives (Section 4.5 / Figure 7).
>
> We quantified this effect on the GPT-5 Judge Subset run by measuring narrative length (number of per-step facts) and comparing it with the evaluator's selections. We find essentially no relationship between narrative length and selection quality; BJudge chooses the longest narrative only 11% of the time, close to the 10% random baseline among 10 candidates, and selected narratives fall at the 36th percentile of candidate length on average. The evaluator therefore does not exhibit a preference for longer or more detailed narratives. We have updated the text to reflect this.
>
> The Code-GUI handoff failures illustrated in Figure 7 are instead driven by an evidence asymmetry. When the coding agent completes a task through a single programmatic edit, the rollout often produces few screenshots and a sparse narrative, leaving limited observable evidence that the task succeeded. An equivalent GUI-based rollout generates many screenshots and a richer narrative, providing substantially more evidence for evaluation. In the observed failure cases, the evaluator favored the evidence-rich GUI rollout even when it failed because the successful code-based rollout provided too little evidence of success. Consistent with this explanation, the correct code rollout's narrative was roughly half as long as the selected rollout on average.
>
> > Broader Impact: the current two-sentence statement is not sufficient; address the risks of scaling autonomous multi-rollout execution against real services.
>
> We have expanded the Impact Statement to address this directly. It now discusses how scaling to many parallel rollouts increases the likelihood of unintended side effects; that effects propagating through external services are not contained by virtualization and may be irreversible (the side effects of unselected rollouts are not undone by selection); and concrete operational safeguards (sandboxed environments with scoped credentials and human confirmation for consequential actions).